# Simultaneous orientation and 3D localization microscopy with a Vortex point spread function

Christiaan N. Hulleman[1,4], Rasmus Ø. Thorsen[1,4], Eugene Kim[2,3], Cees Dekker [2], Sjoerd Stallinga [1,4✉] & Bernd Rieger [1,4✉]

Estimating the orientation and 3D position of rotationally constrained emitters with localization microscopy typically requires polarization splitting or a large engineered Point Spread Function (PSF). Here we utilize a compact modified PSF for single molecule emitter imaging to estimate simultaneously the 3D position, dipole orientation, and degree of rotational constraint from a single 2D image. We use an affordable and commonly available phase plate, normally used for STED microscopy in the excitation light path, to alter the PSF in the emission light path. This resulting Vortex PSF does not require polarization splitting and has a compact PSF size, making it easy to implement and combine with localization microscopy techniques. In addition to a vectorial PSF fitting routine we calibrate for field-dependent aberrations which enables orientation and position estimation within 30% of the Cramér-Rao bound limit over a 66 μm field of view. We demonstrate this technique on reorienting single molecules adhered to the cover slip, λ-DNA with DNA intercalators using binding-activated localization microscopy, and we reveal periodicity on intertwined structures on supercoiled DNA.

[1] Department of Imaging Physics, Delft University of Technology, Delft, The Netherlands. [2] Department of Bionanoscience, Kavli Institute of Nanoscience, Delft University of Technology, Delft, The Netherlands. [3] Max Planck Institute of Biophysics, 60438 Frankfurt, Germany. [4] These authors contributed equally: Christiaan N. Hulleman, Rasmus Ø. Thorsen, Sjoerd Stallinga, Bernd Rieger. ✉email: s.stallinga@tudelft.nl; b.rieger@tudelft.nl

Single-Molecule Localization Microscopy (SMLM), with flavors like (F)PALM[1,2], (d)STORM[3,4], and (DNA)-PAINT[5,6], have made nanoscale structural information beyond the diffraction limit more easily accessible to biologists. These super-resolution techniques commonly focus on localizing single emitters in the two dimensions of the focal plane, and sometimes in the third dimension along the optical axis. The role of molecular orientation in localization can often be ignored as the fluorescent labels are flexibly attached to the biomolecule of interest and can rotate or wobble freely, thereby appearing as isotropic emitters. In case the fluorophores are more rigidly attached, emitter orientation is either a nuisance for estimating the position of the emitters accurately or can give access to the anisotropy of the underlying biological structure. Imaging the orientation of constrained fluorescent labels has been used to visualize changes of fibroblasts under treatment[7], to reveal the underlying orientation of amyloid fibrils[8], and to track motor proteins[9,10]. Besides biological applications, the orientational information can also be used to visualize nanoscale deformations in material sciences[11].

Visualization of emission patterns from fixed fluorescent emitters dates back to near-field studies[12] and studies with high-NA fluorescence microscopes, leading to the observation of ring-shaped spots originating from molecules oriented along the optical axis[13]. Localization of these rotationally fixed molecules with a standard 2D Gaussian model leads to inaccuracies due to their dipole emission patterns[14]. The localization accuracy of these fixed dipoles is significantly worse in the presence of aberrations[15], and with a small amount of defocus errors can amount to 100 nm[16]. The impact of rotational diffusion has been studied in ref. [17] in which the authors show that a localization bias on the order of 10 nm already occurs when the fluorophores are constrained to a cone half-angle of 60°. This localization bias could be avoided altogether by removing the radially polarized component[18,19], regrettably doing so reduces the number of signal photons. Alternatively, the polarization could be split into an $x$ and $y$ component and the $y$ and $x$ position could then be fitted from each respective polarization channel[20]. This also avoids the localization bias but results in an asymmetric localization precision.

A polarized standard Point Spread Function (PSF)[20,21] cannot only be used to avoid localization bias but also to identify the in-plane orientation of fixed emitters. Unfortunately, it is very difficult to determine the out-of-plane orientation from an in-focus polarized PSF as the PSF shape does not vary a lot as a function of the polar angle. When defocusing fixed emitters with a standard PSF, the observed pattern varies more as a function of orientation, opening up a way to estimate the full orientation from a single defocused image[22–24]. A defocus of up to 1 μm spreads the emitted photons over many pixels, which has the drawback of adversely affecting localization precision. By interleaving in-focus localization with defocused spot fitting, a compromise between orientation estimation and localization precision can be made[10]. For a limited set of orientations the angles can even be extracted from in focus single-molecule images in case of sufficiently high Signal-to-Noise Ratio (SNR)[11,25,26].

An alternative way to estimate orientation and position comes from engineering the Point Spread Function (PSF). Here techniques modify the emission beam phase profile[27–30] in the pupil plane and split the polarization[31] to measure different polarization components on the camera separately. The effective size of these engineered PSFs is large, ranging from 4 to 16 times the Rayleigh criterion in size ($R = 0.61\lambda/NA$) in each polarization channel. This splitting of photons over a large area and multiple polarization channels reduces the Signal-to-Background Ratio (SBR)[21,32] and limits the density of emitters per frame for localization microscopy. A recently published method referred to as Coordinate and Height super-resolution Imaging with Dithering and Orientation (CHIDO)[33] overcomes this problem with a more compact PSF that encodes the orientation and position into two different polarization channels. It turns out that this method performs 2–5 times worse on simulations than predicted by theory, despite a good maximal theoretical precision in both 3D location and orientation. Furthermore the experimental precision is even 5–6 times worse than the theoretical precision, which could be explained by the mismatch between their fitting model and experimental data. This method requires many additional optical components in the emission path such as polarization splitting and a custom-produced stress-engineered optic. In addition, the latter component is particularly difficult to align and calibrate.

In this work we overcome these drawbacks by introducing the Vortex PSF, a PSF engineering approach to simultaneously determine molecular orientation and position in all three spatial dimensions that does not require different polarization components to be imaged separately. Furthermore the Vortex PSF enables access to the degree of rotational mobility so that the flexibility of binding of the fluorophores can be probed. Both the azimuthal and polar angle precision are good with no ambiguity between ±45° azimuthal angles, and the lateral localization precision is close to that of a non-engineered PSF. This is because a single imaged spot with the Vortex PSF has a footprint of only 4–6 times the Rayleigh criterion in size, compared to a standard non-engineered PSF that has a width of 2–4 times the Rayleigh criterion. Therefore, depending on the emitter's orientation, the Vortex PSF is only 1.5–2 times larger than a standard non-engineered PSF. This relatively small spot footprint on the camera leads to a favorable SBR, high localization precision, and a sparsity constraint for localization microscopy on the same order of magnitude as a standard (non-engineered) PSF. The use of a calibrated aberration map over the entire Field-of-View (FOV) with a fully vectorial PSF model in the parameter estimation avoids aberration-induced biases and successfully reaches the Cramér–Rao Lower Bound (CRLB). This makes it possible to maintain precision and accuracy across standard FOVs of tens of μm. Implementation of the Vortex PSF into an existing setup is easy and can be realized via an affordable off-the-shelf component. We showcase the method by tracking orientational jumps of single molecules on a cover slip and imaging $\lambda$-DNA using Binding-Activated Localization Microscopy (BALM)[34,35]. Furthermore the Vortex PSF enables the identification of periodicity in synthetically supercoiled DNA structures.

## Results

**Vortex PSF concept.** The orientation of a constrained dipole emitter is characterized by three numbers, the in-plane azimuthal angle $\phi$, the polar angle with respect to the optical axis $\theta$, and a parameter quantifying the degree of rotational mobility $g_2$ (Fig. 1a). The $g_2$ parameter is a weight parameter quantifying the contribution of the fixed versus free dipole component to the overall PSF, which is sufficient to quantify the impact of orientational constraint, irrespective of the form of the constraint, e.g., "wobble-in-cone" or harmonic orientational potential well[36]. In the completely freely rotating emitter case $g_2 = 0$ and for a fully constrained emitter $g_2 = 1$. The angles $\phi$ and $\theta$ describing the dipole orientation lie on a hemisphere; $0 \leq \theta < 180°$ and $0 \leq \phi < 180°$.

A fixed dipole emitter that is oriented perpendicular to the optical axis ($\theta = 90°$) emits fluorescence that is in-phase throughout the Fourier plane of the emission path. When imaged without the Vortex PSF, this emitter yields Gaussian-like spots on the camera due to constructive interference in the center. The

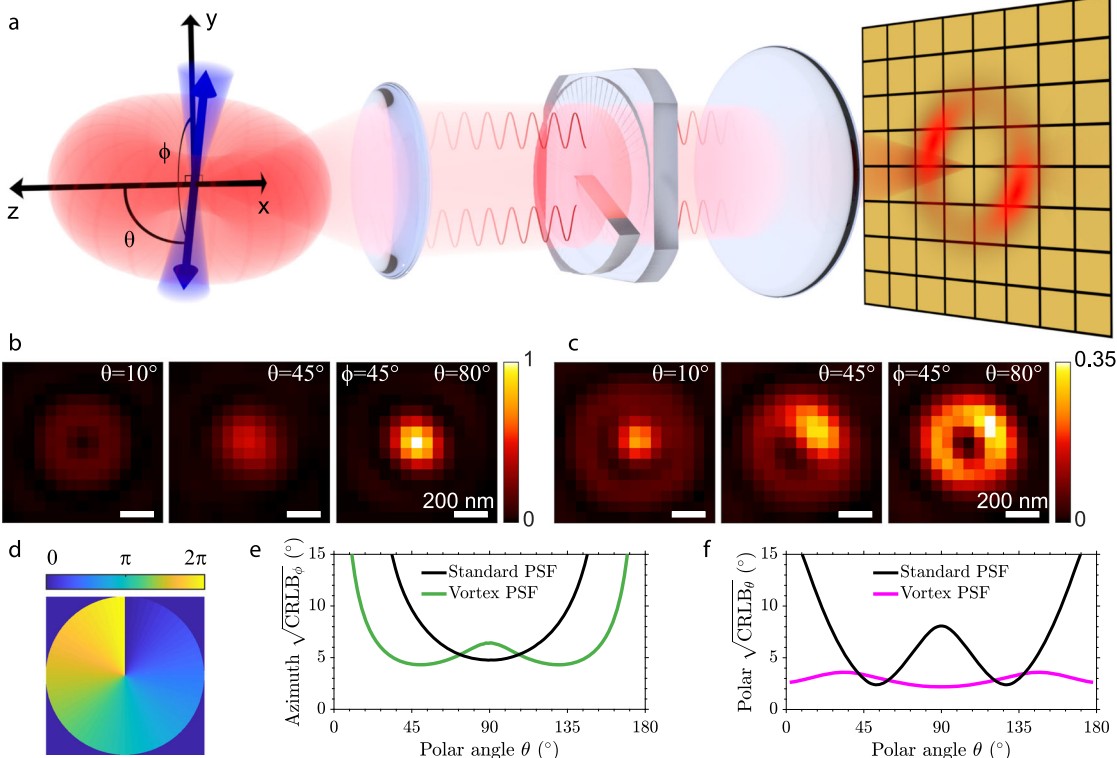

**Fig. 1 Vortex PSF concept. a** A constrained dipole emitter is defined by the polar angle $\theta$ and azimuthal angle $\phi$. The blue cone represents the degree of rotational constraint ($g_2$) of the dipole emitter and the red torus-like shape represents the dipole emission. The microscope, equipped with a vortex phase plate, induces a radial $\pi$ phase difference in the Fourier plane of the emission path. This results in an asymmetric donut-like shape for fixed emitters on the camera which we call the Vortex PSF. **b** Simulated standard PSF of fixed emitters ($g_2 = 1$) with polar angles from left to right (10, 45, 80 degrees) and an azimuthal angle of 45 degrees. **c** Simulated Vortex PSF with the same angles and rotational constraint. **d** Phase profile of the vortex phase plate. **e** Azimuthal angle CRLB from simulated images as a function of the emitter polar angle (4000 signal photons, 10 background photons per pixel and $g_2 = 1$). **f** Polar angle CRLB as a function of the emitter polar angle with the same SBR and rotational constraint.

emission from a dipole oriented along the optical axis ($\theta = 0°$) captured by a high-NA objective has two regions on opposite sides of the Fourier plane of the emission path that are out of phase with respect to each other. Without the Vortex PSF this emitter yields a ring-shaped PSF with a zero in the middle due to destructive interference. Figure 1b shows simulated images of standard PSFs with nearly in and out-of-plane orientations ($\theta = 80°$ and $\theta = 10°$). The slight asymmetry in the PSF arising from the azimuthal orientation ($\phi = 45°$), however, is difficult to identify by the eye.

The Vortex PSF can be realized as an addition to any standard fluorescence microscope and only requires a single spiral phase plate in the Fourier plane of the emission path (Fig. 1a). This phase plate is commonly used in STED microscopy[37] to alter the excitation PSF; here we use it to engineer the emission PSF instead. The phase plate consists of a phase vortex of topological charge 1, thus the phase delay is a single spiral ramp from 0 to $2\pi$, where radially opposing points always have a $\pi$ phase difference between them (Fig. 1d). Adding a vortex phase plate in the Fourier plane of the emission path inverts the phase relationship described earlier for a standard PSF and creates what we call the Vortex PSF. Now out-of-plane ($\theta = 0°$) orientations have constructive interference in the center, generating a central spot surrounded by a dark ring and an additional dim ring. In-plane ($\theta = 90°$) orientations have destructive interference in the center resulting in a zero surrounded by a bright ring. Due to the polarization and directional emission from the fixed dipole emitter, the intensity distribution changes along this ring as a function of the azimuthal angle. When varying the polar angle

from $\theta = 0°$ to $\theta = 90°$ the central bright spot moves outwards asymmetrically, distinctly changing the PSF shape as a function of the polar angle. Simulated Vortex PSF shapes for polar angles ($\theta = 10°$, $\theta = 45°$, $\theta = 80°$) indicate a substantial change as a function of the polar angle, as well as a clearly recognizable impact of the azimuthal angle ($\phi = 45°$) on spot shape (Fig. 1c).

Fitting molecular dipole orientations using the standard in-focus PSF is difficult because of its symmetries. The PSF is almost rotationally symmetric for all polar angles except around $\theta = 90°$, where there is a slight asymmetry of the PSF as the spot is wider in one direction than the other. The azimuthal precision, quantified by the CRLB, is indeed worse for all polar angles except near $\theta = 90°$ (Fig. 1e). Furthermore, the standard PSF is also symmetric around the polar angles $\theta = 0°$ and $\theta = 90°$ yielding an unfavorable precision in estimating the polar angle around these angles (Fig. 1f). These symmetries are broken by the Vortex PSF, resulting in a good precision over all possible orientations (Fig. 1e, f). Of course, the precision of the azimuthal angle is still expected to diverge to infinity for polar angles approaching $\theta = 0°$ and $\theta = 180°$ as the azimuthal angle is undefined when the dipole is aligned along the optical axis. Note that due to the symmetry of the dipole it is sufficient to use half the unit sphere to uniquely define the dipole angle.

**Simulated precision and accuracy.** We have tested the vectorial Vortex PSF model with extensive simulations to predict the experimental conditions under which the Vortex PSF can be used to correctly estimate the parameters $\Theta = (x, y, z, N, b, \phi, \theta, g_2)$

(see Supplementary Note 1 for a detailed description of the model). The parameters are as follows: $x, y, z$ are the emitter coordinates, $N$ the number of signal photons, $b$ the number of background photons per pixel, and orientation parameters $\phi, \theta, g_2$ as described earlier (Fig. 1a). We found that all model parameters can be estimated with precision at the CRLB for all molecular orientations and degrees of orientational constraint (Supplementary Fig. 1), provided the signal-to-background ratio ($SBR = N/b$) is sufficiently high. A practical lower limit, assuming 10 background photons per pixel (typical for SMLM), is around $SBR \geq 200$ (Supplementary Fig. 2). At higher signal photon counts the required SBR is less stringent, for example with 4000 signal photons an $SBR \geq 40$ ($b \leq 100$) is sufficient.

For dipole emitters oriented uniformly on unit sphere with $g_2 = 0.75$ and $SBR = 4000/10$ the parameters can be estimated with a localization precision of $\sigma_{xy} = 5.6$ nm, $\sigma_z = 27$ nm and orientation precision of $\sigma_\phi = 5.5°$, $\sigma_\theta = 3.1°$, and $\sigma_{g_2} = 0.08$ (Supplementary Fig. 1). The polar precision appears almost constant over the unit sphere, whereas the azimuth precision performs well within polar angles ($20° < \theta < 160°$) after diverging for emitters oriented along the optical axis. Such polar range is notably broader than the standard PSF, as shown in Fig. 1e. The amount of rotational diffusion of an emitter affects the possible orientational and axial precision while it does not affect the lateral precision. The orientation can be estimated to a precision within $\sigma_\theta < \sigma_\phi < 10°$ given a rotational diffusion $g_2 > 0.4$ (<58° cone half-angle). Outside this range the PSF becomes too smeared out, and the orientation information is mostly lost. The optimal axial performance is for fixed emitters $\sigma_z = 23$ nm ($g_2 = 1$), whereas it worsens for freely rotating emitters up to $\sigma_z = 49$ nm ($g_2 = 0$). In this case, when the emitters are freely rotating, the Vortex PSF has a slightly worse lateral precision compared to a non-engineered PSF ($\sigma_{xy} = 5.9$ nm versus $\sigma_{xy} = 4.5$ nm). Generally the estimation for all parameters works well over a $z$-range of ±300 nm with a region of interest (ROI) of $15 \times 15$ pixels around an emitter with a pixel size of 65 nm (Supplementary Fig. 1). This relatively small ROI is ideal for dense single-molecule localization microscopy and attains a reliable parameter estimate for $|z| \leq 300$ nm. In Supplementary Fig. 3 we compare the performance to standard defocus-based orientation fitting over a $z$-range of ±1000 nm in simulation. The ROI size for this comparison is increased to $31 \times 31$ pixels. These simulations show that the precision with the Vortex PSF is better than with the standard PSF for $z, \phi$, and $g_2$ around focus $|z| \leq 300$ nm and the polar angle ($\theta$) is better in an even wider region $|z| \leq 600$ nm. Outside this region the performance is equivalent. With the fluorophores in an aqueous medium ($n = 1.333$) the difference is less pronounced as the spherical aberrations from the refractive index mismatch break the symmetry around focus but the Vortex PSF still performs better for $|z| \leq 300$ nm. In these refractive index mismatched conditions the usable $z$-range with the Vortex PSF is about twice the range for standard defocused orientation fitting.

Simulations show furthermore that optical aberrations must be taken into account in the fitting model. Unknown or inaccurately calibrated aberrations with values deviating from the actual values by more than 36 m$\lambda$ affect the imaging model such that the estimator introduces biases and no longer reaches the CRLB (Supplementary Fig. 4). The aberration modes astigmatism and coma notably degrade the azimuthal precision by a factor of ~2, and additionally, astigmatism also degrades the localization precision by a factor of ~2. However, when the model is well calibrated with these aberrations, the estimator reaches the CRLB with no biases. To avoid such inadequate estimation and variation over a large field of view, we have developed an aberration map based on the Nodal Aberration Theory (NAT)[38]

(See Methods and Supplementary Note 2). Using this aberration map we obtain reliable results over a large FOV despite significant changes in aberrations throughout the FOV (Supplementary Fig. 5).

**Proof-of-concept**. To first verify the Vortex PSF's functionality and performance, we have imaged ATTO 565 embedded in a thin layer of PMMA (Polymethylmethacrylate) (see the methods section for sample preparation and imaging protocol for any samples mentioned). Figure 2 shows the results of one of these experiments. Figure 2a shows through-focus images of a single molecule in PMMA acquired with the vortex phase plate together with the fitted PSF model and its estimated parameters. Note that a bigger region-of-interest (ROI) $31 \times 31$ is used to verify the model over a large $z$ range compared to typical localization, where $15 \times 15$ is recommended. The same single molecule was imaged in quick succession without the vortex phase plate to verify the estimated orientations. As it is challenging to estimate orientations with the standard PSF on a single frame with high precision, the entire z-stack is used to retrieve a single estimate, which is used as a ground-truth measure. This estimate differentiates slightly from defocus imaging as described in the literature[10,22,23], where only a single focal slice is used. For the single molecule in Fig. 2a, its orientation found for the standard PSF z-stack fit ($\phi, \theta$) = (48°, 61°) agrees well with the angles found for the Vortex PSF z-slice fits ($\phi, \theta$) = (49 ± 1.5°, 61 ± 1.7°). Here the Vortex PSF uncertainty is the standard deviation estimate of 11 focal slices, corresponding to a dynamic range of 1000 nm. Following the same procedure as for the molecule in Fig. 2a, the estimated orientation of 21 different molecules is depicted in (Supplementary Fig. 6). The mean deviation between the orientation found with the standard PSF and Vortex PSF is ($\Delta\phi, \Delta\theta$) = (−0.4 ± 1.4°, −0.2 ± 1.2°), indicating no bias between the two imaging modes. The Vortex PSF's mean precision is ($\sigma_\phi, \sigma_\theta$) = (2.3°, 1.8°) which is, respectively, only 28% and 20% larger than the estimated lower limit ($\sqrt{CRLB_\phi}, \sqrt{CRLB_\theta}$) = (1.8°, 1.5°). For the 21 molecules, the signal level varies one order of magnitude ($4.4 \times 10^3 - 45 \times 10^3$, with a mean of $17 \times 10^3 \pm 12 \times 10^3$, and a mean background of 19 photons per pixel). The mean rotational constraint parameter is $g_2 = 0.86$, with a standard deviation of 0.03. This corresponds to an average wobble cone semi-angle of $\alpha = 25.3°$ which is similar to the previously found rotational constraint of single molecules in PMMA $\tilde{\gamma} = 0.85$[39] ($\alpha = 27.8°$) and $\tilde{\gamma} \approx 0.8$[30] ($\alpha = 30.7°$).

The estimated $z$-position shift between frames of the molecule in Fig. 2a is $\Delta\tilde{z} = 97 \pm 11$ nm, which matches the piezo shifts of 100 nm. The relationship between the estimated $z$-position and the piezo position averaged over 37 molecules is fitted with a linear function, resulting in a slope of $-0.99 \pm 0.01$ and has a Root Mean Square Error (RMSE) of 16 nm over a 1200 nm range as shown in Fig. 2b. To further show the quality of fit, cross-sections in the $x-z$ and $y-z$ planes are shown in Fig. 2c, d. The agreement between the experimental data and fit with the vectorial Vortex PSF model is generally excellent. A striking detail is that even the fringe details away from focus match well. The lateral localization error, measured on individual $z$-slices between the two estimation modes, is 5 nm and 4 nm (RMSE) in the $x$ and $y$ direction. These validation experiments show that the orientation of fixed dipole emitters and their 3D position can be reliably estimated from individual focal-slice Vortex PSF images.

**Re-orientation dynamics**. The Vortex PSF can track dynamic changes in the orientation of single molecules on a cover slip. We

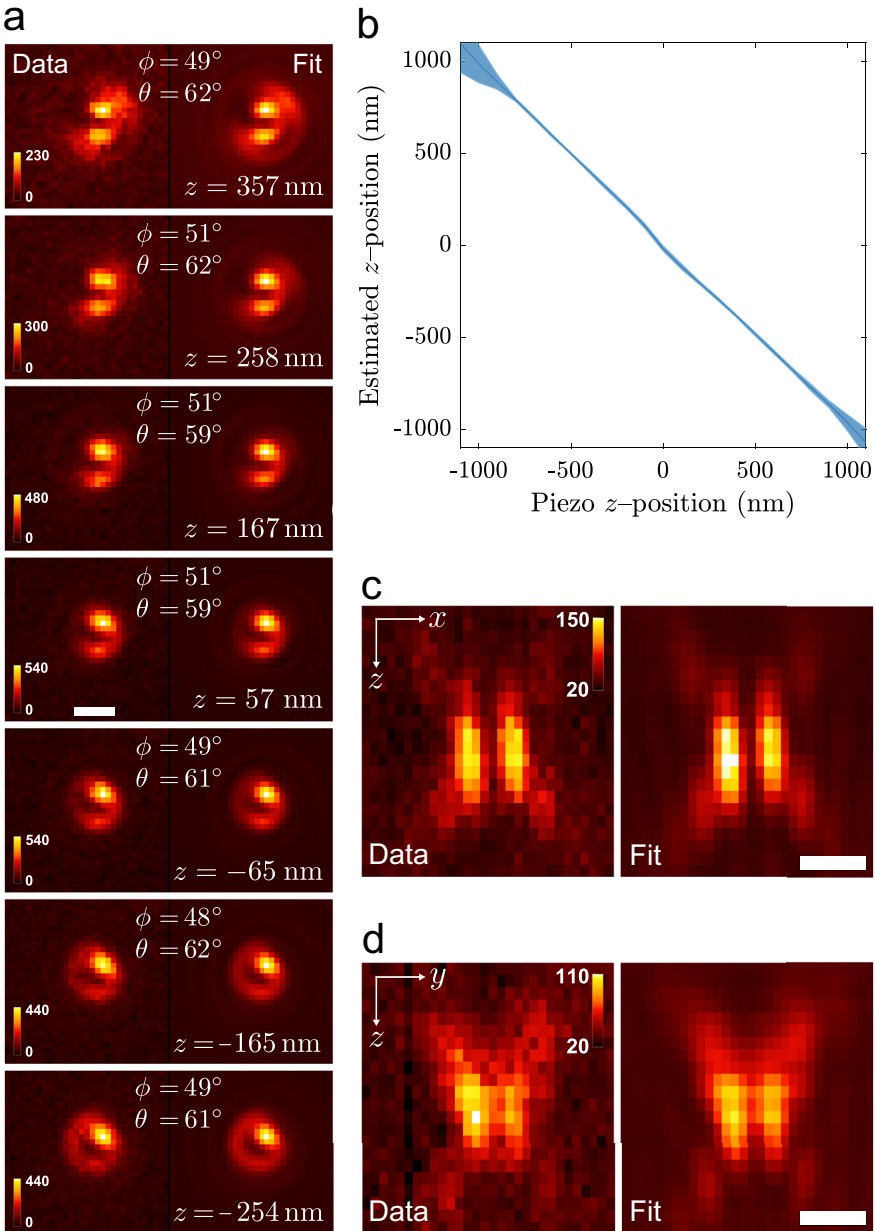

**Fig. 2 Vortex PSF validation. a** Vortex PSF model fitted to an experimental $z$-stack of one single molecule with its estimated parameters listed in each frame. All sub-image pairs are contrast stretched with the same factor for better visibility of the PSF shape and the color bars are in units of photons per pixel. The mean signal count in the frames is $25 \times 10^3$ with a standard deviation of $4 \times 10^3$ and a mean background of 24 counts per pixel. The mean degree of orientational constraint of this molecule is $g_2 = 0.92$ with standard deviation of 0.02. **b** Mean estimated $z$ position as a function of the piezo $z$ position with the shaded region representing ± one standard deviation. The average is taken from 37 single molecules of varying orientations where the piezo $z$ is realigned in processing to account for the in-focus position not corresponding exactly with $z_{piezo} = 0$. The slope is negative due to the opposing definition of $z_{piezo}$ and the sample $z$. **c** The cross-section in $x-z$ and **d** $y-z$ of a Vortex PSF, with the measurement left and fit right (pixels are stretched proportionally in $z$, color bar units: photons per pixel). The estimated signal photon counts were in the range $4 \times 10^3 - 50 \times 10^3$, and the estimated background photon counts were in the range $10-40$ photons per pixel. All scale bars are 500 nm.

have observed that a large portion of out-of-plane oriented ATTO 565 single molecules directly spin-coated on glass (without PMMA) show re-orientation when followed over time, indicating metastable adhesion to the glass surface (see Supplementary Movie 1). Figure 3a shows the Vortex PSF of the three highlighted molecules from Supplementary Movie 1 that re-orient over time. This can mainly be seen from the dark region that shifts location between the bright central spot and the ring around it. Additional time traces of various fitting parameters are shown in Fig. 3b. The

three different molecules that undergo these re-orientation events seem to primarily make jumps in the azimuthal angle. The re-orientations are also observed in the rotational constraint $g_2$ as it takes a lower value in the transition frames, which indicate a more freely rotating molecule or a superposition of orientations before and after the transition. These types of re-orientation would lead to large position biases in standard localization microscopy, which are avoided with the Vortex PSF (Fig. 3c). There does appear to be a marginal position shift as the molecules

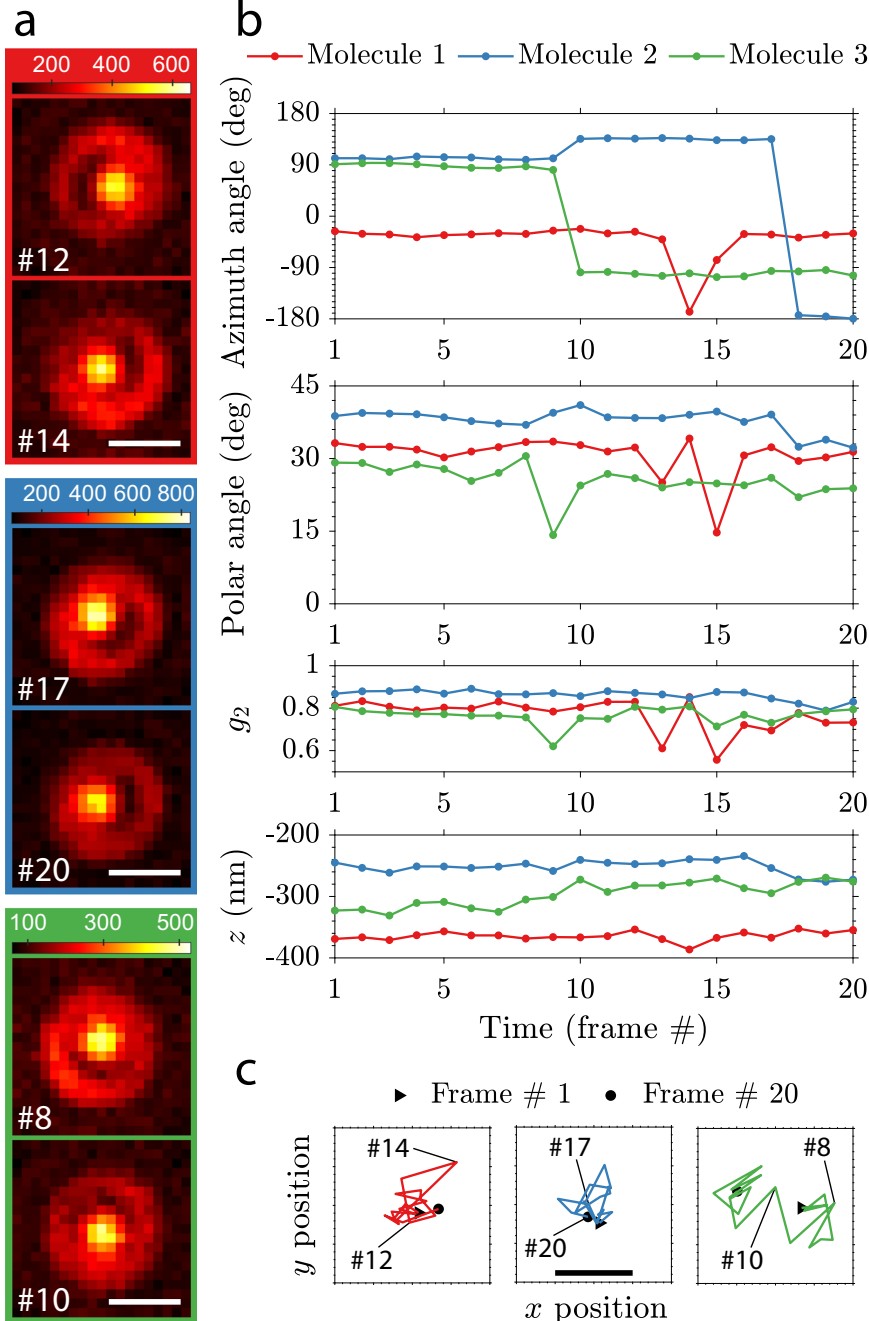

**Fig. 3 Re-orientation dynamics of single molecules imaged with the Vortex PSF. a** Raw Vortex PSF images of 3 different molecules undergoing orientational transitions with frame numbers indicated in the bottom left (900 ms exposure, scale bar 500 nm and color bar units: photons per pixel). Many more re-orienting molecules from the same acquisition can be seen in Supplementary Video 1 and the experiment was repeated with similar results. **b** Estimated parameters over 20 frames of the three different molecules showing orientational transitions. The photon count per frame (mean ± one standard deviation) is: $N_1 = 4.7 \times 10^4 \pm 0.2 \times 10^4$, $N_2 = 5.9 \times 10^4 \pm 1.0 \times 10^4$, $N_3 = 3.3 \times 10^4 \pm 0.5 \times 10^4$ and background is: $b_1 = 91 \pm 2.0$, $b_2 = 89 \pm 1.1$, $b_3 = 80 \pm 1.7$. **c** Lateral localization of the emitters undergoing re-orientation with 1 nm ticks, a scale bar of 10 nm and the raw frames from **a** indicated. The localizations are drift corrected with the average trajectory of 10 stationary molecules (3.6 nm in $x$, 4.2 nm in $y$, and 19 nm in $z$).

re-orient; however, the average standard deviation of the localizations $\sigma_{xy} = 2.7$ nm and the $\sqrt{\mathrm{CRLB}_{xy}} = 1.7$ nm show that the residual shift is small compared to the theoretical precision limit.

**Super-resolved $\lambda$-DNA.** A wide variety of biological structures can be labeled with rotationally constrained fluorophores[9,10,21,40]. We chose to demonstrate our technique on frequently studied $\lambda$-DNA labeled with DNA intercalators[34,35,41] that transiently bind

between the base pairs. The molecular dipole moment of the DNA intercalators is typically oriented perpendicular to the DNA axis[39,42,43], making this an ideal test case. We chose to use Sytox Orange which is believed to be a mono-intercalator[44] and further confirmed by elongation measurements matching that of mono-intercalators[41] (the exact chemical structure is undisclosed by the manufacturer). The sparsity required for localization microscopy is inherent from the transiently binding Sytox Orange that is essentially non-fluorescent when not intercalated[44]. In order to

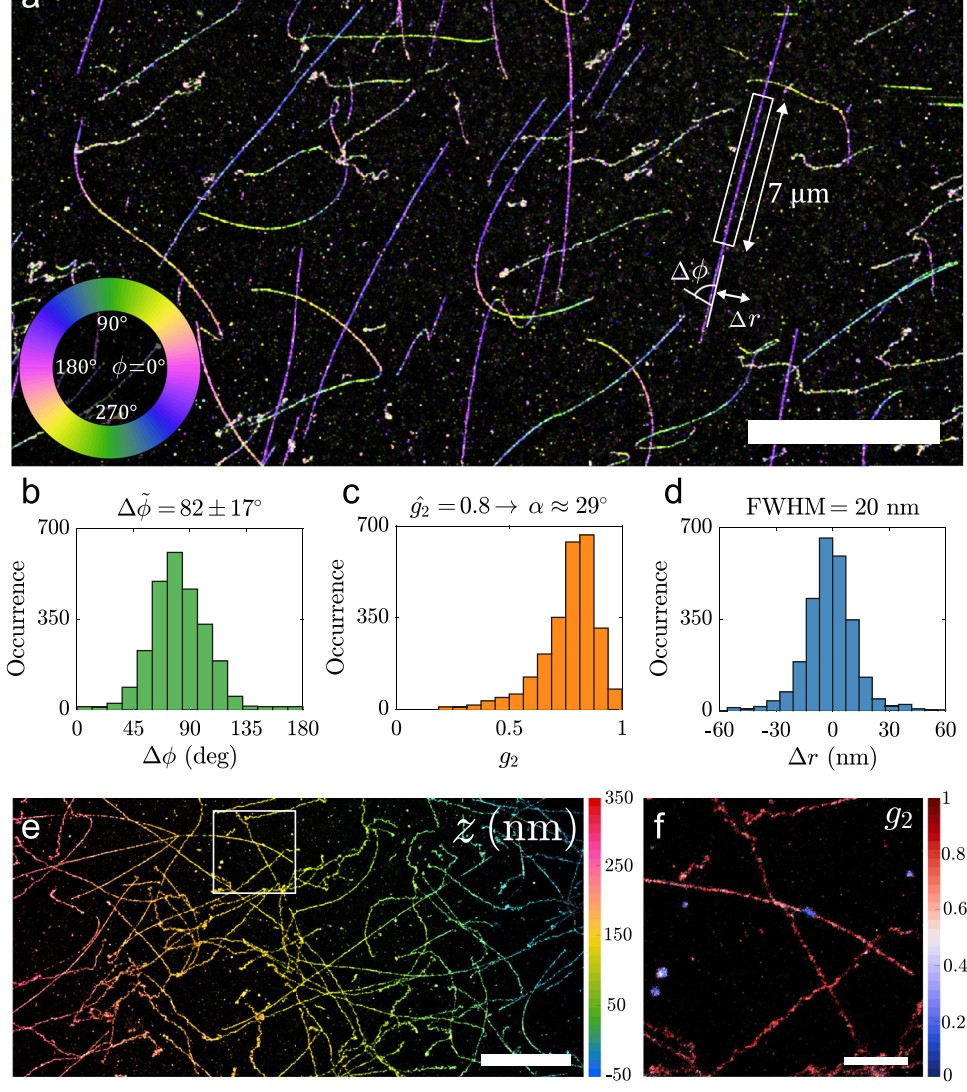

**Fig. 4 Super-resolution image of λ-DNA. a** λ-DNA colorized as a function of the azimuthal angle. **b** Relative azimuthal angle histogram ($\Delta\phi$) with respect to the DNA axis from the strand highlighted in **a** ($\Delta\phi = 82°$, MAD($\Delta\phi$) = 17°). **c** Distribution of $g_2$ from the same strand, the fitted peak $g_2 = 0.8$ corresponds to a wobble cone semi-angle $\alpha = 29°$. **d** Position deviation from the spline fit to the DNA axis with a Full-Width Half-Maximum (FWHM) of 20 nm (Filtered to $35° \leq \theta \leq 145°$). **e** Tilted λ-DNA sample colorized as a function of z-position. **f** λ-DNA from highlighted region in **e**, colorized as a function of $g_2$. Scale bars are **a** 10 μm, **e** 5 μm, and **f** 1 μm. These experiments were repeated with similar results ($N > 10$).

visualize the λ-DNA in a fluorescence microscope, molecular combing is used to align and stretch the DNA on a cover slip by a receding water–air interface[45,46].

Figure 4 shows the potential of combining 3D localization microscopy and orientation estimation by imaging λ-DNA. The super-resolution reconstruction can be color-coded with one of the orientation parameters (azimuthal angle, polar angle or degree of orientational constraint) or the z-position. Figure 4a shows a subsection of the entire FOV with an azimuthal angle color-coding. The in-plane molecular orientation is clearly perpendicular to the orientation of the DNA-strands. The local orientation of the DNA strand can be determined by fitting a spline curve to the localizations on the strand. The angle difference ($\Delta\phi$) between the fluorophore and DNA axis orientation can be estimated. In other words $\Delta\phi$ describes the azimuthal angle in a coordinate frame where the DNA strand is locally pointing in the $x$-direction. Analyzing a single strand shows a mean azimuthal angle difference between the fluorophore and the DNA axis of $\widetilde{\Delta\phi} = 82°$ with a median absolute deviation (MAD)

of 17° (Fig. 4b). This is essentially the same as found before ($\Delta\phi = 87°$ and MAD($\Delta\phi$) = 18°)[39] and similar to measurements with YOYO[42,43]. The degree of orientational constraint $g_2$ along the strand is estimated with a peak at $g_2 = 0.8$, which corresponds to a maximum semi-angle $\alpha = 29°$ in the framework of the wobble-in-cone model (Fig. 4c), slightly larger than ~22° found previously[39,42]. All these parameters are estimated while attaining a lateral resolution (20 nm FWHM λ-DNA line-width) typical for BALM with dimeric dyes[34,35] (Fig. 4d).

PSF fitting methods are sensitive to knowing the experimental parameters (like NA, refractive index etc.) and the Vortex PSF is no exception. The influence of variations of aberrations across the FOV has been investigated and mitigated by creating an aberration map for our microscope (see discussion in section 2.2 and Methods). Spherical aberration can, however, vary easily from sample to sample depending on the refractive index, thickness of the cover slip, and the setting of the correction collar. Pushing the biases of our method to levels below the precision requires caution when using catalogue values. E.g. using the

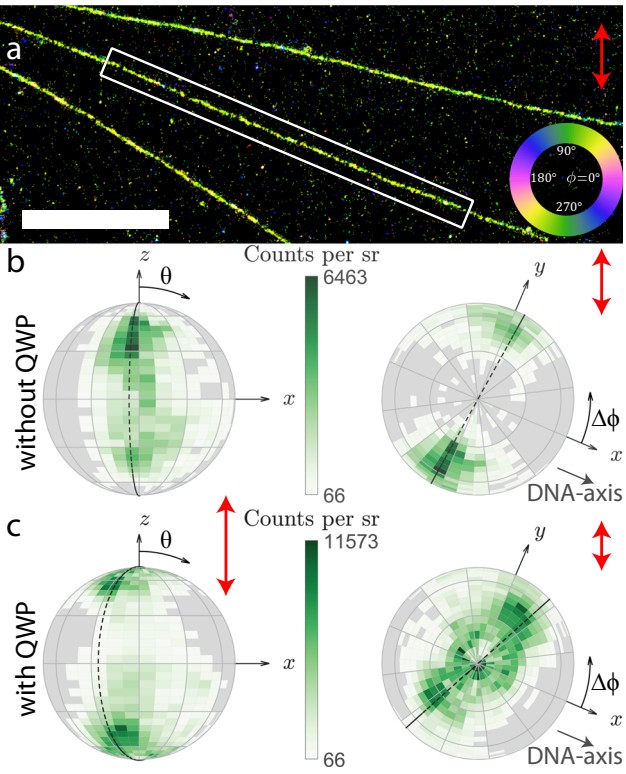

**Fig. 5 Correlation between orientation parameters of λ-DNA under different excitation polarization. a** Super-resolution image of λ-DNA strands color coded with the same azimuthal map as Fig. 4a (scale bar is 3 μm). **b** Relative dipole orientations with respect to the DNA axis aligned along the x direction, illuminated without the QWP. On the left the view is aligned along the y axis, on the right is a top view aligned in the negative z direction. The dashed line highlights the primary population around Δϕ = 84°. **c** Relative dipole orientations of the same λ-DNA strand with a QWP in the illumination path. The dashed line highlights an additional population around Δϕ = 65°. Each bin spans 5° in the polar direction and 10° in the azimuthal direction, additionally there is a 30° coarse grid. Red arrows indicate the excitation polarization intensity. The distributions are similar on many different DNA-strands (N > 20) and samples (N > 10).

comparably free dipole value $g_2 = 0.3$, as opposed to the nearly fixed dipole value $g_2 = 0.8$ of the DNA strands. The achieved precision from these experiments, determined from repeated localizations of multiple on-events from the same emitter (Supplementary Fig. 10), is 5.4 nm and 29.7 nm in the lateral and axial dimension, and the azimuth and polar angle precision are 6.0° and 3.9° with an orientational constraint precision of 0.06. The localization and orientation precision values determined in this way are within the estimated error bars of the CRLB, with the photon distribution showing a median signal of 4600 photons and a median background of 10 photons per pixel (Supplementary Fig. 10).

The orientational binding landscape can be further investigated by analyzing correlations between the orientational parameters ($θ$, $ϕ$, $g_2$), combined with illuminating with different excitation polarizations. To that end we have imaged the same region both with and without a quarter waveplate (QWP) in the illumination laser path in Total Internal Reflection Fluorescence (TIRF) conditions (Fig. 5a). Without the QWP the excitation polarization is s-polarized (polarization extinction ratio of 240:1 and an excitation polarization orientation of $θ = 90°$ and $ϕ = 90°$), most effectively exciting molecules around $θ = 90°$ and $ϕ = 90°$ (Fig. 5b). With a QWP the polarization is less in-plane and more out-of-plane resulting in less selective excitation (polarization extinction ratio of 3:1 and a primary excitation polarization orientation of $θ = 15°$ and $ϕ = 90°$). This results in a polar distribution with more localizations around $θ = ±40°$ (Fig. 5c). Without the QWP we find the expected relative angle of $Δϕ = 87°$ between the fluorophore and DNA axes with almost no dependence on the polar angle (Fig. 5b). With the addition of the QWP a correlation between the polar and azimuthal angle becomes visible. Next to the population of molecules with close to in-plane orientations, that still have the expected relative angle $Δϕ = 84°$, a second population of molecules with more out-of-plane angles $θ = ±40°$ appears, that has a relative angle of $Δϕ = 65°$ in Fig. 5c. These correlations between the orientational parameters are only found in the DNA experiments and are independent of the DNA orientation (Supplementary Fig. 11). Control experiments on fixed single molecules in air and PMMA show a uniform distribution over the azimuthal angles with no correlation to polar angle or degree of orientational constraint (Supplementary Fig. 12). Supplementary Table 1 summarizes the localization density of individual analyzed DNA strands for all relevant figures.

One would expect a uniform polar angle distribution for free DNA strands as Sytox Orange should intercalate in any orientation, averaging out any base pair selectivity along the strand. The proximity of the cover slip to the DNA strand could limit the physical space available for intercalators, thereby creating a non-uniform polar angle distribution and possibly shift the equilibrium azimuthal angle away from the orientation perpendicular to the DNA strand. However, this does not explain the preference for $Δϕ = 65°$ over $Δϕ = 115°$ for both polar angle regions. This may originate from the helical structure of the DNA and the binding potentials between the intercalator and the DNA.

A different hypothesis for the observed correlations between the orientational parameters is a change of the DNA structure to S-DNA due to overstretching. Previously it was found that overstretching results in a change of azimuthal angle to $Δϕ = 54°$[42], comparable to the value we find. The typical length of λ-DNA strands in our datasets is 17.4 μm which is 7.4% longer than its crystallographic length[47]. Although this corresponds to a relatively low percentage of overstretching it is possible that the binding affinity is not as low in the proximity of the cover slip as for free DNA strands, resulting in a relatively large population of tilted orientations compared to the DNA strand. We did not

catalogue value for the refractive index of the cover slip ($n = 1.523$) leads to a bias of 8.1 nm between localizations with $θ < π/2$ and $θ > π/2$ (Supplementary Fig. 7). Using an optimized cover slip refractive index of $n = 1.5209$ this bias is reduced to just 0.8 nm. Without the fine drift correction and optimized refractive index setting the FWHM of localizations of a λ-DNA strand would be 35 nm. Furthermore if the calibrated field-dependent aberrations were not taken into account in the fitting model, the localization distribution would become non-Gaussian with a λ-DNA line-width that is even twice broader (73 nm FWHM, see Supplementary Fig. 8).

To demonstrate the ability to resolve the lateral structure independent of defocus we use an intentionally tilted λ-DNA sample. This sample is shown in Fig. 4e, where the estimated z position reveals the slope of the tilted λ-DNA sample. Comparison of the slope of the tilted and non-tilted data set (Supplementary Fig. 9) accurately retrieves the additional tilt of 0.42° (0.4° applied). The z localizations have a variation of 64 nm FWHM around the plane of the cover slip which is to be expected from a mean $\sqrt{\mathrm{CRLB}_z}$ of 25 nm. Figure 4f shows color-coding with the degree of orientational constraint $g_2$ that can be used to identify patches where binding to partially detached DNA strands and/or non-specific binding occurs. These are visible with a

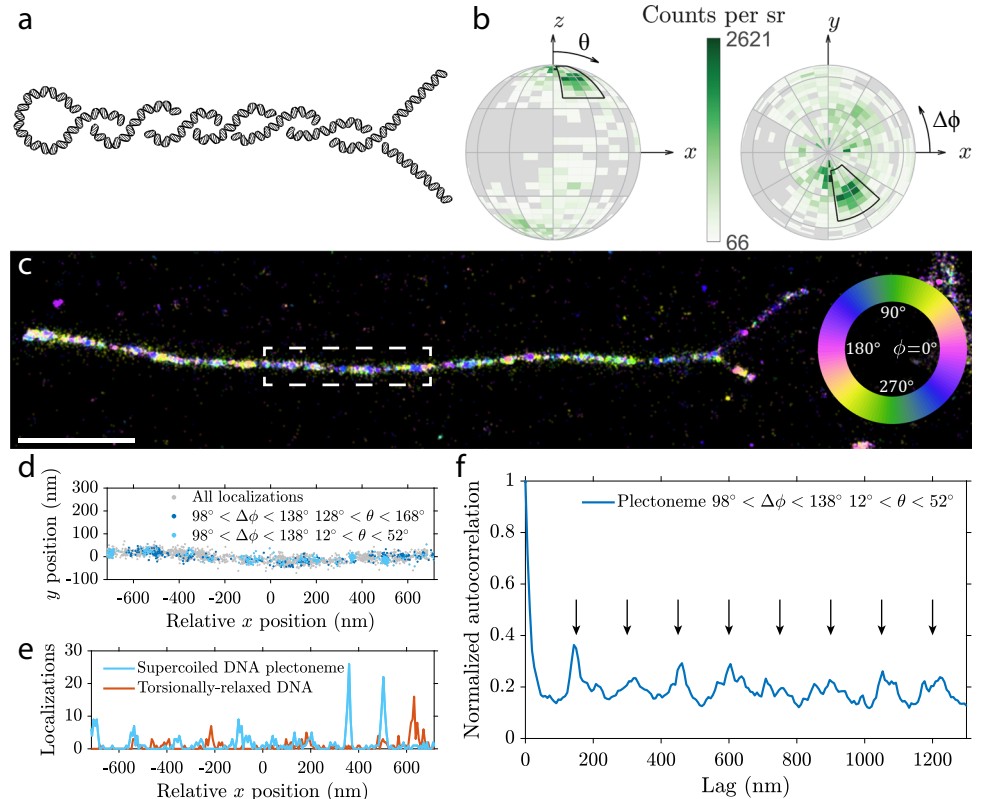

**Fig. 6 Orientation and spatial correlation in a supercoiled DNA plectoneme. a** Diagram of a plectoneme formed by a single supercoiled DNA molecule where two strands are intertwined upon applied torsion. **b** Orientation distribution of a typical individual strand before overlap, with the DNA axis aligned in the $x$ direction. The primary orientation analyzed in further subfigures is highlighted with the black region. Each bin spans 5° in the polar direction and 10° in the azimuthal direction, additionally there is a 30° coarse grid. **c** Azimuthal orientation of dipoles on a plectoneme (rotated 90 degrees clockwise to fit figure, scale bar 1 µm). Other plectonemes show similar orientations as seen in Supplementary Figs. 13, 14 and in an additional sample. **d** Localizations in different angular subsets from the plectoneme highlighted in **c**, showing periodic clusters of localizations. **e** Localizations from **d** arranged in 6.5 nm bins along the $x$-direction compared to a torsionally relaxed DNA molecule. **f** Autocorrelation of localizations along the plectoneme in 6.5 nm bins. The equidistant arrows mark periodic peaks in the autocorrelation function, indicating a periodicity of ~150 nm.

observe a correlation between the orientational parameters and the position along the DNA strand, that would correspond to domains along the strand with different orientational binding. This makes this hypothesis less likely.

**Supercoiled DNA**. With the help of the Vortex PSF we can investigate local orientation of twisted structures at the super-resolution level. Here we image supercoiled DNA, where external torsion is applied to the double-stranded DNA to create a larger scale coiled structure. Such a twisted DNA can form a so-called plectoneme where two helices intertwine each other, similar to an old telephone cable, as illustrated in Fig. 6a. Plectonemes were generated in a 42-kbp DNA molecule that was tethered at both its ends to a cover slip, and then aligned to the surface by buffer flow application, and attached to the glass slide for imaging. While the imaging conditions, dye, and buffer were the same as for the $\lambda$-DNA sample, the sample preparation method was altered significantly: to induce supercoiling, the DNA strands were initially attached to the cover slip at the both ends in the absence of Sytox Orange. Subsequently, Sytox Orange was introduced which locally unwinds DNA and generates torsional stresses in the torsionally constrained DNA molecule, thus generating supercoils on DNA[48]. Shortly after, the flow of buffer containing a low concentration of formaldehyde aligned the supercoiled DNA in one direction and constrained the DNA to the cover slip surface

(more detailed sample preparation protocol can be found in the methods).

Orientation estimation of intercalated emitters with the Vortex PSF reveals various interesting properties of supercoiled DNA structures. Firstly the primary dipole orientation is different compared to the $\lambda$-DNA case where no additional torsion was applied. Figure 6b shows the orientation distribution of a supercoiled strand before it twists around itself. The highlighted peak around $\Delta\phi = 298°$ and $\theta = 32°$ (equivalent to $\Delta\phi = 118°$ and $\theta = 148°$) occurs on various strands. A 180° rotationally symmetric population also occurs on other DNA strands that are presumably oriented in the opposite direction with a peak around $\Delta\phi = 118°$ and $\theta = 32°$ (data not shown). This different azimuthal orientation found in the supercoiled DNA sample could be due to the different surface attachment protocol. This could change the physical space or electrostatic potential around the intercalator and DNA molecule.

Secondly, and more interestingly is that there appears to be a periodicity in certain orientations along the plectoneme. In Fig. 6c the azimuthal angles in the range of $90° < \Delta\phi < 135°$ shown in green to blue seem to occur periodically along the plectoneme. Figure 6d shows the individual localizations in different orientation subsets identified earlier along the region highlighted in Fig. 6c. This periodic spacing is particularly clear in the subset of $98° < \Delta\phi < 138°$ and $12° < \theta < 52°$ shown in light blue, representing ~10% of all localizations. The localizations are

binned in 6.5 nm intervals along the $x$-direction, such that an autocorrelation of the localization density can be calculated. The periodic pattern from Fig. 6d is more evident in the binned localizations in Fig. 6e, compared to binned localizations from a torsionally relaxed DNA molecule (from Supplementary Fig. 14b) where no periodicity can be clearly identified. The autocorrelation along the entire plectoneme from Fig. 6c is shown in Fig. 6f. The autocorrelation contains periodic peaks which is indicative of a periodic function. These periodic peaks are highlighted by equidistant black arrows and occur ~150 nm apart. Two other plectonemes shown in Supplementary Fig. 13 have a periodicity of ~122 nm and ~130 nm. A few of the periodic peaks found in these orientational subsets can be identified in the full-data autocorrelation but cannot be directly identified as there are too many aperiodic peaks in the full-data autocorrelation. This periodicity was not found on DNA sections from the same dataset that are presumably torsionally relaxed as they are only tethered to the cover slip on one side (Supplementary Fig. 14). Additionally this periodicity is also not found in the $\lambda$-DNA datasets (Supplementary Fig. 15).

The length of the supercoiled DNA molecule estimated from localizations is $12.6 \pm 0.4$ μm (Supplementary Table 2). Compared to the crystallographic length the estimated length is 10% shorter, which may be attributed to the intertwining of the DNA molecule as the straight line projection is an underestimate of the actual DNA molecule length. The ratio between localization density on the coiled section and the individual strands before crossover is $1.94 \pm 0.15$ (Supplementary Table 2). This doubling of the localization density confirms there are two DNA strands in the coiled section. A significant change in the binding affinity in either of the sections could indicate a change in DNA conformation[42]. In future, it will be of interest to carry out a systematic study on many more plectonemes.

Lastly the orientation of supercoiled sections before and after the plectoneme appear to have opposing orientation shifts. The peak relative orientation in the region to the top right of a plectoneme is $\Delta\phi = 111°$, $\theta = 36°$ and to the bottom right is $\Delta\phi = -102°$, $\theta = 29°$, with the full distribution of all 3 sections of a plectoneme shown in Supplementary Fig. 16. These two orientations appear shifted in opposite directions and are not 180 degree rotation symmetric which would be expected from a non-supercoiled strand.

Aside from the discernible plectoneme shape in the lateral localizations, there is further evidence of a difference between the supercoiled DNA and non-coiled $\lambda$-DNA. One difference is the change in the preferential binding orientations between the two datasets. Another is the apparent periodicity in localizations of certain orientations along the supercoiled DNA sections. Interestingly the periodicity of these localizations is on the same size scale as the expected superhelix pitch[49].

## Discussion

Using the Vortex PSF we achieve accurate parameter estimation, avoiding the large position biases of several 10s of nm commonly seen with fixed dipole emitters in localization microscopy. Instead we reduce these biases to below the localization precision by matching experimental parameters like the refractive index, if one uses the nominal values for these quantities we find only a small bias of up to several nm. A key ingredient for our overall high accuracy is that we use calibrated field-dependent aberrations and supercritical angle fluorescence in the vectorial PSF model to avoid additional model mismatches. The relatively compact spot shape enables a more favorable trade-off between the precision of estimating the position and orientational parameters compared to a large PSF footprint. The vectorial PSF estimator achieves a

precision close to the CRLB, and the CRLB for the Vortex PSF is relatively uniform for all possible emitter orientations. The SBR requirement of at least 200 (2000 signal photons and 10 background photons) can be met in typical SMLM experiments, and allows relatively short exposures of ~30 ms to be used.

To put the achievable precision of the Vortex PSF into perspective we have made a comparison to state-of-the-art PSF engineering methods for orientation estimation in Supplementary Table 3. Compared to the Tri-spot PSF[30], the Vortex PSF has a similar performance in the azimuthal angle, but the polar precision is significantly better. Methods focused on polarization splitting, for example a polarized PSF[21] or CHIDO[33], benefit from a better azimuthal precision as distribution of photons between the two polarization channels reveals more information about the polarization of light and consequently the orientation of the emitter dipole moment. This also appears to yield a slightly better precision in the wobble cone semi-angle $\alpha$ for methods with polarization splitting. Overall there is not a lot of variation in the lateral localization precision as all but the Tri-spot PSF are fairly compact PSFs. It is important to realize that currently most methods except the Vortex PSF and CHIDO cannot estimate the full 3D position along with both orientation angles and the rotational constraint. Comparing the only two methods that could estimate all these parameters simultaneously, we see that CHIDO has the advantage that the polarization splitting improves the azimuthal and axial precision. The estimator used in CHIDO, however, only gets to within 200–500% of the theoretical CRLB limit on simulated PSFs. On experimental data this is even worse 500–600% ($\sigma_{xy} = 13$ nm compared to $\sqrt{\mathrm{CRLB}_{xy}} = 2.3$ nm, $\sigma_z = 50$ nm compared to $\sqrt{\mathrm{CRLB}_z} = 7.5$ nm, $\sigma_\theta = 5°$ compared to $\sqrt{\mathrm{CRLB}_\theta} = 0.8°$, solid angle $\sigma_\Omega = 0.9$ compared to $\sqrt{\mathrm{CRLB}_\Omega} = 0.13$). This could be caused by aberrations in their system or by difficulties in characterizing the birefringence of their stress-engineered optical element. The estimation of parameters with the Vortex PSF reaches the CRLB in simulation over a wide range of signal to background ratios as shown in Supplementary Fig. 2. On experimental data we have shown that the Vortex PSF achieves a precision no worse than 30% above the CRLB on fixed molecules (section 2.3) and close to the CRLB on intercalators attached to DNA (Supplementary Fig. 10). Finally, the Vortex PSF has the virtue of simplicity in an optical setup compared to the Tri-spot PSF and CHIDO, which both use a custom phase-mask and polarization splitting. Summarizing, the polar precision is better with the Vortex PSF compared to other compact PSF designs, whereas the azimuthal precision is slightly worse compared to more experimentally complex polarization splitting methods. The Vortex PSF could be expanded to utilize polarization splitting and potentially improve both the azimuthal and axial precision. This would come at the cost of the simplicity of the optical setup.

We have visualized and measured orientational transitions of single molecules with metastable attachment to a glass surface. Furthermore we have applied our method to $\lambda$-DNA, corroborating previous findings that the azimuthal angle of the intercalator dipoles is almost perpendicular to the DNA axis. Our method uncovered a preferential polar orientation of intercalators attached to $\lambda$-DNA on a cover slip along with a correlation between the orientational parameters. The Vortex PSF also unveiled a periodicity along plectonemes on supercoiled DNA molecules which could be indicative of its supercoil periodicity.

We have applied the Vortex PSF to these cases to illustrate its functionality, but in principle, the Vortex PSF can be applied to any sparse sample of constrained dipole emitters. Combined with a sparsity inducing single-molecule localization microscopy technique, super-resolution images can be complemented with

orientation information to differentiate various sub-sets in the data, such as identifying different binding modes, different orientational configurations or local deformations. Nanoscale interactions could be investigated using chemical models on the single-molecule scale.

An interesting question to address in future studies would be to compare the Vortex PSF to fundamental (quantum) limits of the estimation of orientational parameters[50,51]. Another step to advance the Vortex PSF concept could be a speed up of the fitting algorithm by developing a GPU implementation, or by using spline interpolated models obtained from PSF calibrations[52] extended to take into account field-dependent aberrations. The feasibility of estimating the orientational confinement that is not rotationally symmetric around the preferential axis defined by the minimum of the orientational potential well in addition to the other parameters could also be investigated. Going from a uni-axially symmetric to a biaxially symmetric orientational con-finement would bring the number of orientational parameters that must be estimated from three to five. Aside from the general degree or rotational constraint and the primary dipole orientation an additional parameter is needed to describe the primary direction of rotational diffusion and another for the degree of asymmetry. A reliable estimation of the then total number of 10 parameters (instead of 8) may require a more complex setup involving e.g., polarization detection in addition to the vortex phase plate. Another intriguing possibility is to study the char-acteristics of bis-intercalators with a double-dipole model. Finally, the analysis could be extended into the regime of slow orienta-tional diffusion. In that regime the illumination polarization has an impact on PSF shape, implying that modulation of the illu-mination polarization into the method could generate useful information on the orientational constraint and diffusion of the molecule.

## Methods

**Fitting model.** We use standard Maximum Likelihood Estimation (MLE) using an image formation model that describes the expected photon count across the image as a function of the molecule position $\mathbf{r}_0 = (x_0, y_0, z_0)$, signal photon count $N$ of the entire PSF on the camera, background photons per pixel $b$, and dipole orientation $\Omega_0 = (\phi_0, \theta_0)$ with the degree of orientational constraint $g_2$, giving a total of 8 parameters. The underlying PSF model is the fully vectorial PSF model, as in earlier work[15,25,53–55], but now extended to estimate the dipole orientation along with the degree of orientational constraint. The image formation model is based on a weighted sum of the freely rotating dipole PSF and the fixed dipole PSF corre-sponding to the equilibrium dipole orientation[36], plus a constant background:

$$H(\mathbf{r}, \Omega) = N\left[\frac{(1-g_2)}{3}H_{\text{free}}(\mathbf{r}) + \frac{g_2}{3}H_{\text{fixed}}(\mathbf{r}, \Omega)\right] + \frac{b}{a^2} \qquad (1)$$

where $a$ is the pixel size and $0 \leq g_2 \leq 1$ represents the degree of orientational constraint, with the limiting cases of a fully free dipole $g_2 = 0$ and a fully fixed dipole $g_2 = 1$. This seems simplistic compared to more complex rotational diffusion models[50,56–58]. The appropriateness of the model for rotational diffusion faster than the fluorescence lifetime, however, is demonstrated in ref. [36]. An asymmetric rotational diffusion model would require 2 additional fitting parameters, raising the total amount of parameters to 10 which we expect is not realistic to fit with <5000 photons. The vectorial PSF model takes supercritical angle fluorescence (SAF)[59,60] into account. Without accounting for SAF, emitters in the proximity of the cover slip could have an additional position bias up to ~10 nm with a medium of water ($n = 1.33$) and ~25 nm in air ($n = 1$). Further details on the image formation model are given in Supplementary Note 1.

**Calibrated field-dependent aberrations.** An important addition to the fitting model that we make is to take into account calibrated aberrations as done previously[54] and extended further to incorporate the field-dependence of aberrations[61]. We improve upon this treatment by modeling the field dependence of the aberration coefficients using the so-called Nodal Aberration Theory (NAT) instead of 2D polynomials of arbitrary order. This approach is valid for optical imaging systems with small field angles (ratio of FOV to focal length), such as telescopes or microscopes, and has been devised by Shack and Thompson[38], and later extended and used in optical design and characterization studies[62–64].

The key prediction of NAT is that the dependence of aberration coefficients on the field coordinates is well approximated by Taylor series of a low order, such that there are specific relations between the coefficients of these series for different aberrations. Such relations exist for example between the Taylor series for the two astigmatic and the two coma aberration coefficients, giving rise to zero aberration loci (two, respectively, one for astigmatism and coma), so-called "nodes" in the FOV. The advantage of NAT in the current context is that the Taylor series fit for the different aberrations is more robust due to the predefined number of coefficients. Quantitative details on the aberration field dependence and the calibration procedure are given in Supplementary Note 2. The aberration maps for 12 Zernike modes, determined from calibration measurements on 429 beads, are shown in Supplementary Fig. 5. The NAT predictions are in excellent agreement for astigmatism and coma, and good for the other aberration modes.

**Simulation setup.** Simulated point spread functions (PSFs) are generated according to the vectorial PSF model described in Supplementary Note 1. The NA is taken to be 1.45, the wavelength 597.5 nm, the refractive index of the imaging medium 1.33, cover slip 1.523, immersion medium 1.518, a pixel size of 65 nm in object space and a region of interest (ROI) of 15 × 15 pixels. Unless stated other-wise, we take 4000 detected signal photons on the camera and 10 background photons per pixel, and we neglect readout noise but add Poisson noise to each image. The number of photons corresponds to the number of photons captured into the NA and thus spread over the entire FOV. The fraction of signal photons captured within the ROI compared to the entire FOV is typically 0.44 and 0.46 for the standard and Vortex PSF, respectively. The simulations are run for 10,000 randomized instances with coordinates taken from a uniform distribution over ± 1 pixel and molecular dipole orientations uniformly distributed on the unit sphere. That is, if $u$ is a uniform random number from the distribution $U[0, 1]$, the simulated angles are taken to be $\phi_0 = \pi u$ and $\theta_0 = \arccos(1 - 2u)$.

**Sample preparation.** Cover slips (22 × 22 mm No. 1.5, Marienfeld-Superior) and microscope slides (Microscope slides, Menzel Gläser Thermo Scientific) are cleaned by sonication in ethanol for 15 min and are then blown dry with nitrogen. All further mentions of cleaned cover slips and microscope slides are cleaned the same way except the cover slips for the λ-DNA samples. The microscope slides for the λ-DNA samples have a 5–10 mm hole drilled in them in advance to make it possible for the imaging medium to be added from above. Lambda DNA (λ-DNA) (Lambda DNA, Thermo Scientific) is aliquoted into 10 μL portions in PCR tubes and stored at −20 °C. Ascorbic acid (Ascorbic acid, Merck) is divided into ~3 mg portions in PCR tubes and the mass written on the tubes and stored at 4 °C. To make a pH 5.5 solution, 1 μL of 400 mM HCl and 600 mM Tris, is diluted in Milli-Q (MQ) water to a pH of 5.5 (approximately 90 mL). Part of a 5 mM stock solution of Sytox Orange (SYTOX Orange Nucleic Acid Stain, Invitrogen) is diluted in TE buffer (Tris-EDTA buffer solution pH 7.4, Supelco) by 4 tenfold steps to 500 nM and stored at 4 °C.

Fixed single-molecule samples are made by sparsely embedding ATTO 565 in a thin layer of PMMA. 100 mg of PMMA (Poly(methyl methacrylate), Sigma-Aldrich) is dissolved in 10 g of Toluene (Toluene, Sigma-Aldrich). ATTO 565 (ATTO 565, Sigma-Aldrich) is diluted in MQ water in 100 fold steps to ~5 μM. The ATTO 565 dilution is further diluted in PMMA/Toluene in 100 fold steps to a ~5 pM concentration. A 20 μL droplet of the mixture is placed on a clean cover slip in the spin-coater and is spun at 3000 RPM for 2 min. Two strips of double-sided tape (Permanent Double Sided Tape, Scotch) are placed ~1.5 cm apart on a cleaned microscope slide and the cover slip is placed on top with the PMMA facing the tape side.

For single molecules without PMMA the same procedure is followed except now ATTO 565 is diluted only in MQ water to a final concentration of ~500 pM. After spin-coating the cover slip is placed, coated side down, on double-sided tape on a microscope slide.

The λ-DNA samples are prepared by dropping a λ-DNA solution onto a rotating silanized cover slip[34,35]. These cover slips are cleaned more extensively by sonication for 1 h each in ethanol, acetone, and then ethanol again. The cleaned cover slips are stored in ethanol. Before silanizing the surface they are removed from the ethanol and blown dry with nitrogen. An individual dry cover slip is then placed in 15 mL of Poly-L-lysine solution (Poly-L-lysine solution 0.01% sterile-filtered, Sigma-Aldrich) for 5 min and slightly shaken 2–3 times. Thereafter the silanized cover slip is rinsed with MQ water and left to dry overnight. A 10 μL λ-DNA aliquot is thawed and 990 μL of the pH 5.5 solution is added. For a silanized cover slip the optimal combing pH appears to be just below pH 5.5[47]. 40 μL of this solution is applied in a drop wise fashion to the silanized cover slip on the spin-coater rotating at 2500 RPM for 30 s. Thereafter the speed is increased to 7000 RPM and 5 mL of MQ water is applied to rinse away non attached DNA and left spinning for 2 min to dry. A square hole is cut into a piece of double-sided tape (64621, Tesa) and placed around the pre-drilled hole in the microscope slide. The cover slip is placed on the tape and pressed down with the λ-DNA side towards the tape. The ascorbic acid is hydrated with TE buffer to a concentration of 200 mM just before the experiment. 25 μL of the ascorbic acid dilution is mixed with 5 μL of 500 nM Sytox Orange and 470 μL of TE buffer. 200 μL of the imaging buffer with a final concentration of 5 nM Sytox Orange and 10 mM ascorbic acid in TE buffer is

added to the sample through the hole in the microscope slide while on the microscope, focusing and imaging starts as soon as possible.

To determine the aberration maps, 180 nm orange bead samples were used. A 1/100 dilution of 180 nm orange beads (PS-Speck Microscope Point Source Kit, ThermoFisher) is made by mixing 10 μL of beads with 990 μL of MQ water. Using 10 μL of this dilution, 7–10 small droplets are placed around the center of the cleaned cover slip and allowed to dry for ~3 h or overnight. Two strips of double-sided tape (Permanent Double Sided Tape, Scotch) are placed ~1.5 cm apart on a cleaned microscope slide and the cover slip is placed on top with the beads facing the tape side. When ready to image a 20–30 μL droplet of the mounting medium (in our case immersion oil) is placed on the edge between the cover slip and microscope slide. The capillary action gradually distributes the mounting medium between the cover slip and the microscope slide. The sample is placed on the microscope when the mounting medium has reached the other side.

The plectonemic supercoiled DNA were prepared as follows. Coilable DNA constructs of 42 kb in length were synthesized via cloning, PCR, and DNA ligation[65]. We used linearized cosmid-I95 plasmid DNA[66] and extended both the ends of the DNA with short DNA fragments containing biotinylated dUTPs. The resulting biotin-labeled DNA molecules were purified with size exclusion chromatography. The DNA was then introduced to a flow cell with a flow rate of 2 μL/min and subsequently immobilized to the surface via streptavidin–biotin linkage. After observing a reasonable density of double-tethered DNA on the surface, imaging buffer (40 mM TRIS-HCl pH 7.5, 50 mM NaCl, 2.5 mM MgCl$_2$, 1 mM DTT, 5% (w/v) D-dextrose, 2 mM Trolox, 40 μg/mL glucose oxidase, 17 μg/mL catalase) containing 250 nM of Sytox Orange was introduced to the flow cell in order to generate positive supercoiled DNA, which can be manifested by the characteristic dynamics of plectonemic supercoiled DNA reported previously[48,67]. To fix the plectoneme containing supercoiled DNA onto the surface, we introduce 0.2% of formaldehyde in the same imaging buffer with 250 nM of Sytox Orange. The buffer was injected with a large flow rate of 40–50 μL/min in order to fully stretch out the plectonemes and avoid formation of DNA clumps before they adhere to the surface, providing a better alignment quality for the angle-resolved super-resolution localization imaging.

**Imaging protocol**. The main component of the optical setup is a standard microscope (Ti-E, Nikon) with a 100x 1.49 NA objective (Supplementary Fig. 17). A 4F relay system consisting of two 100 mm lenses (AC508-100-A-ML, Thorlabs) relays the original image plane of the microscope to the camera ((Zyla 4.2 PLUS, Andor) or (ORCA-Flash4.0 V2, Hamamatsu) for the supercoiled data) and a vortex phase plate (V-593-10-1, vortex photonics) is placed in the back focal plane between the two lenses. The vortex phase plate is mounted on a small XYZ stage (CXYZ05/M, Thorlabs) for alignment and a kinematic stage (KB75/M, Thorlabs) for quick removal and placement. For microscope control NIS-Elements was used and for image acquisition either Andor Solis or HCImage was used.

The Vortex PSF should be used with a relatively narrow emission bandwidth up to ~60 nm. The design wavelength of the phase plate does not need to exactly match the emission peak as long as both parameters are known and set correctly in the estimator. For fixed dipole emission PSF fitting, the fluorophore must have a single molecular dipole moment; this makes bis-intercalators like TOTO[68] and YOYO[69] unsuitable. These dimeric fluorophores have two transition dipole moments between which the excitation energy can hop[70], resulting in emission from either of the two transition dipoles almost perpendicular to one another.

In the proof-of-principle experiments ATTO 565 is embedded in PMMA, where the polymer immobilizes the fluorophores and has a refractive index of $n = 1.49$ close to that of immersion oil ($n = 1.518$). Two z-stack acquisitions with a 100 nm step-size, 600 ms exposure, and 300 W/cm$^2$ epi-illumination are taken in quick succession with and without the vortex phase plate to compare the Vortex PSF to defocused orientation fitting.

To create a sparse sample of individual dipole emitters that are less constrained and can reorient over time, a low concentration of ATTO 565 without PMMA is spin-coated onto a cover slip. Initially the region is imaged with epi-illumination which bleaches the in-plane molecules which showed almost no re-orientation. The remaining single molecules are excited with 3 kW/cm$^2$ in TIRF conditions as the z-component of the TIRF field more effectively excites out-of-plane molecules. A relatively long exposure time of 900 ms is used to yield raw data with a very high SNR.

For the λ-DNA, after applying a fresh batch of imaging buffer and focusing, 20,000 frames are acquired with a single frame exposure time of 100 ms. The sample is illuminated with 3 kW/cm$^2$ circularly polarized total internal reflection excitation. This results in an excitation profile that is approximately half in-plane and half out-of-plane, and a reduced background due to the limited penetration depth.

**Vortex phase plate alignment**. The vortex phase plate or spiral phase plate is the same phase profile used to create high-quality donut-shaped depletion and excitation profiles from Gaussian laser beams for STED[37] and MINFLUX[71]. Coincidentally this is also the lowest order component of a Double-helix PSF[72], and a discretized version of a spiral phase plate with only 3-4 phase-steps could be used to generate a rotating PSF[73]. Here we use a vortex phase plate with a single spiral from 0 to 2π in 64 steps (V-593-10-1, vortex photonics).

A 1:1 optical relay is built on the emission path of the microscope to place the vortex phase plate in a plane conjugate to the back focal plane of the objective as illustrated in Supplementary Fig. 17. The phase plate is placed roughly halfway between the two relay lenses. To align the vortex phase plate, defocused images are taken of 1 μm beads (TetraSpeck Fluorescent Microspheres Size Kit, ThermoFisher) and diagonally opposing regions are recorded (Supplementary Fig. 18a). If the vortex phase plate is aligned properly, all the PSFs should have the same shape throughout the FOV as shown in Supplementary Fig. 18b. When the vortex phase plate is not in the correct axial position the PSF will vary over the field of view. This is because light from different areas in the sample does not pass through the vortex phase plate in the same place as it is not conjugate to the back focal plane of the objective. The main parts of the PSF to observe for alignment is the peak in the center that does not move when the phase plate is moved and a ring that moves with the phase plate. The process of aligning the vortex phase plate involves moving the rings to be centered over the peaks over the entire FOV. The vortex phase plate should be translated along the optical axis until all the beads look the same throughout the field of view (same offset between the peak and the ring throughout the FOV) (Supplementary Fig. 18c, d). Thereafter the vortex phase plate can be shifted horizontally and vertically so the dark spot overlaps with the center of the bead (Supplementary Fig. 18e, f). Lastly the beads are translated along the optical axis and defocused in the opposite direction to verify that the rings also overlap with the center of the bead there. If the rotation direction of the vortex phase plate is unknown ($K(\rho) = \beta/(2\pi)$ or $K(\rho) = -\beta/(2\pi)$), then fixed single molecules can be fitted with both orientations and the correct setting will have a visually better fit (should be especially evident on molecules with $\theta = \pm 45°$).

**Data analysis**. The acquired images are offset and gain corrected to convert analog-to-digital units (ADUs) into photon numbers[74]. Then, candidate pixels with a single molecule signal are identified using an intensity threshold typically chosen as the background plus a constant of around 10 photons. These candidate pixels are segmented into ROIs of size 15 × 15 pixels centered at the local centroid. The local centroid gives a better first estimate of the emitter's position than the local maximum due to the Vortex PSF shape. These ROIs are fitted with a vectorial PSF model using Maximum Likelihood estimation (MLE). The vectorial PSF model incorporates emitter parameters described in Supplementary Note 1 and is further tailored to take field-dependent aberrations into account described in Supplementary Note 2. The optical parameters and experimental settings are set to match the experimental setup in Supplementary Fig. 17.

In the fitting routine potential outliers and poor localizations are filtered out as follows. The number of iterations in the MLE is terminated if the relative difference between successive log-likelihood values in the iteration drops below $10^{-6}$ or if the maximum number of iterations, which is set to 30, is reached. Typically fitting converges within 20 iterations, events are rejected if the maximum number is reached without convergence. Moreover, when the estimated molecule position is more than 3 pixels away from the center pixel in the ROI, it is rejected. Finally, converged localizations are tested with a normalized chi-squared ($\chi^2$) test[54]. If a localization has a $\chi^2$ value outside the region $0.75 \leq \chi^2 \leq 3$, it is also rejected. Additional filtering is specified where used.

The resulting localizations are corrected for sample drift following the method of Schnitzbauer et al.[75], implemented in the Picasso software (v0.2.8), available at github.com/jungmannlab/picasso. The slow timescale lateral drift for the λ-DNA experiments was on the order of 1.5 pixels (~100 nm) over 30 min. Additionally a fine drift correction is applied to this dataset, utilizing straight λ-DNA sections and minimizing the deviations from a spline fit to the DNA axis over time. Supplementary Figure 19 shows that these deviations of two different strands with the same orientation are correlated and can be used to improve the drift correction. For this dataset 27 strands of varying orientations are analyzed and the $\Delta r$ deviations are projected into $x$ and $y$ deviations. These $x$ and $y$ deviations are then averaged with a weighting prioritizing good localization precision and a high number of localizations to generate the fine drift trajectory. This reduces the residual drift from a standard deviation of 7.6 nm (Supplementary Fig. 19a) to 3.7 nm (Supplementary Fig. 19b). Subsequently localizations of the same emitter during the emitter's on-time are linked under the condition that the position and orientation estimate between subsequent localizations is less than 3 times the largest uncertainty as shown in Supplementary Fig. 10.

All images are rendered with a Gaussian blurring using the scripts from the INSPR toolbox[76]. The images in Fig. 4 are rendered with a super-resolution pixel size of 6.5 nm, the image in Fig. 5 with a pixel size of 20 nm, and the image in Fig. 6 with a pixel size of 13 nm.

Estimating the axes of single DNA strands is performed by fitting a spline to the localization data. First, a DNA strand is selected from the localization data incorporating all localizations. A spline curve is fitted to the localizations using MATLAB's built-in function *fit()* employing a smoothing spline with a smoothing parameter of $10^{-1}$. In order to determine the azimuthal orientations of molecules with respect to the DNA axis, the shortest distance between a given molecule and a point on the spline curve is determined, and the tangent line to the point on the spline is calculated using finite differences, giving the local DNA-strand orientation.

**Reporting summary**. Further information on research design is available in the Nature Research Reporting Summary linked to this article.

## Data availability

The raw image files and processed localization data generated in this study have been deposited in the 4TU database[77] with the following https://doi.org/10.4121/13031837.

## Code availability

Matlab code for Vortex PSF simulation, fitting, and aberration calibration is available on github.com/imphys/vecfitcpu_vortex and the 4TU database[78].

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

## Acknowledgements

This work was supported by the European Research Council ERC (648580) and the National Institutes of Health NIH (U01EB021238) and C.D. acknowledges support from ERC Advanced Grant 883684. We thank Srividya Ganapathy for help preparing the pH 5.5 solution and Marijn Siemons for help with the vectorial PSF estimator applied to SAF conditions.

## Author contributions

C.N.H. conceived the project. S.S and B.R. supervised the research. R.Ø.T., C.N.H., and S.S. developed the PSF estimator. R.Ø.T. developed the NAT aberration estimator and performed all simulations. C.N.H. performed all experiments. E.K. and C.D. devised the plectoneme sample preparation method and E.K. prepared the supercoiled samples. R.Ø.T. analyzed the data. C.N.H. and R.Ø.T. wrote the manuscript, S.S., E.K., C.D., and B.R. edited it.

## Competing interests

The authors declare no competing interests.
