## [Peer Review File · Nature Communications]

REVIEWER COMMENTS

Reviewer #1 (Remarks to the Author):

In their manuscript, Rieger and coworkers present a very cool application of the vortex phase plate in single-molecule localisation microscopy: it allows engineering PSFs such that not only location in X, Y, and Z can be determined, but that also the orientation for the molecule (theta, phi, and ψ) can be determined, with high accuracy and up to the statistical limit. Overall, the explanation, the physical foundation and the demonstrations are very sound and clear, also for the broader Nature Comm audience. I think this is a very interesting new tool for the single-molecule community, it appears quite easy to implement in many setups and the authors provide detailed information for this including fitting routines. I wholeheartedly recommend publication in Nature Comm. I would like to make a few comments that might help the authors to make their manuscript even better.

Science:

- The authors show that a SBR of at least 200 is required. This is pretty high, but that is of course not surprising, 6 parameters should be extracted from a single image. Could the authors discuss (in the end?) this a bit more, describing what this would practically mean and to what limitations this leads (e.g. time resolution / bleaching etc. in the DNA intercalator experiments).
- I think it is / could be a bit confusing that the authors use the same theta / phi for the DNA experiments and for the other experiments (and theory), without explicitly mentioning it. For the DNA, the DNA axis is taken as the central axis, for the other experiments it is the optical axis (which is perpendicular to the DNA axis). I mean, this is fair enough, but maybe the reader would be helped to explicitly remark this. A further question is what this "coordinate transformation" does to the errors in the angles that are determined.
- The lambda DNA experiments. The authors hypothesise that the heterogeneity they observe might be to a fraction of SDNA in their combed DNA molecules. This could indeed be the case. It would be very, very cool to test this in optical tweezers / fluorescence experiments, but that is out of the scope of this paper (but I would recommend the authors to do this in a follow up). Maybe the authors could consider (in the current manuscript) other explanations / effect: could the SDNA be dynamically / locally formed and disappear (and thus be less location specific)? Might there be other effects due to torsionally constraining the DNA and inducing supercoiling due to intercalator binding?

Text:

- Abstract: "Developed an engineered..." Don't the authors use a widely used (for STED) and commercially available device in another way than before, opening up a great new application? To me these beginning words do not really describe what the authors did.
- Line 61: "Instead of only avoiding..." I did not understand this sentence and how it connects to sentences before and after.

Review of *Nature Communications* manuscript NCOMMS-20-43805

“Simultaneous orientation and 3D localization microscopy with a Vortex point spread function”

In this work, Hulleman, et al., develop an engineered point spread function (PSF), called the Vortex PSF, to simultaneously measure the dipole orientation, 3D position, and degree of rotational constraint of single-molecule emitters. A vortex phase plate inserted the microscope’s back focal plane (BFP) changes the image of a single molecule into a donut-like shape. The vortex PSF has superior performance for estimating the 3D position and orientation of single molecules compared to the standard PSF, as quantified by the Cramér-Rao bound (CRB). It is also able to discern re-orientation dynamics of single Atto 565 molecules spin-coated on glass. Finally, the authors demonstrate imaging the positions and orientations of DNA-intercalating dyes on λ -DNA. They find an interesting correlation between the polar angle of the dye versus its binding rigidity to the DNA.

Overall, the authors present an intriguing new method for adding dye orientation information to single-molecule localization microscopy (SMLM). The technique is a clever reuse of an existing PSF, and the authors carefully characterize their technology both in theory and in experiments. However, the long-term significance of the method, as demonstrated, is still unclear. Additional comparative evidence to existing states of the art should be added before a final decision can be reached. Detailed recommendations for improvements are below.

Major comments

1. The suitability of the vortex PSF for various SMLM applications is a little unclear from this work. In particular, the performance of the vortex PSF (estimation precision, predicted signal-to-noise ratio (SNR), etc.) is quantitatively compared to only the standard PSF. The lack of other comparisons is problematic because, as the authors state, the standard PSF is only suited for estimating the orientations of very bright fluorophores or fluorophores with certain orientations.
 - a. Please make the comparison in Fig. 1(e,f) more robust by computing the orientation estimation precision (CRB) of some combination of the following:
 - i. A modestly defocused standard PSF, which is very easy to implement. There should be some intermediate defocus values that enable robust localization and orientation measurement, similar to Refs. [20-24]. Please comment on if the Vortex PSF is uniformly better than all possible defocused standard PSFs, or if the standard PSF has superior performance for some set of defocus values and orientations.
 - ii. A polarized standard PSF (as used in Ref. [46]), which only needs a polarizing beamsplitter or a dual-channel polarization module like the Opto-Split from Cairn. Ref. [46] showed that both in-plane orientation and rotational mobility

g_2 could be robustly measured for lower photon counts than reported here. Please comment on the strengths and weaknesses of the Vortex PSF compared to this technique.

- iii. CHIDO (as used in Ref. [32]). From Ref. [32], it seems that CHIDO is much more precise than the Vortex PSF for the same number of photons detected. Please comment on the strengths and weaknesses of the Vortex PSF compared to this technique.
- iv. The analyses in the following preprint seem to be relevant. Is the Vortex PSF superior to any of the methods mentioned?

Single-molecule orientation localization microscopy II: a performance comparison

Oumeng Zhang, Matthew D. Lew
arXiv:2010.04064

- b. Lines 362-363: The statement “the CRLB for the Vortex PSF is good for all possible emitter orientations” does not currently have adequate quantitative evidence to support it. The comparisons requested above are very important to include, and this statement should be revised to reflect the new observations.
2. Several of the experimental demonstrations need additional quantification/analysis to support the conclusions of this work.
- a. Please report the mean and standard deviation of g_2 estimates from the example molecule in Fig. 2 and the 21 molecules in Supplementary Fig. 5. Are these measurements consistent with molecules confined within PMMA? If there are discrepancies, please discuss possible causes for them.
 - b. Please include raw images of molecule 2 and molecule 3 before and after their orientation “jumps” in Fig. 3 (i.e., frame 9 for molecule 2 and frame 18 for molecule 3). Please add an annotation to the various trajectories in Fig. 3(c) showing the position of the 3 molecules when the orientation “jump” occurred. Is there any correlation between the orientation and position trajectories of each molecule?
 - c. Please clarify the meaning of and rationale for reporting Δz in Fig. 3(c). It seems to me that it would be more useful to report here the absolute z position of each molecule with respect to the focal plane.
 - d. Fig. 4(e) reports the z positions of labeled λ -DNA without any quantification or analysis. What is the measured tilt angle? Is this degree of tilt expected given the apparatus used in the experiment? Does the sample appear to be perfectly planar, i.e., flat? What is the variation in measured z positions, and how much do they deviate from the CRB? Overall,

I find this axial localization demonstration to be weak, and a more compelling demonstration would improve the paper.

- e. Please comment on the non-Gaussian distribution of the measured $\Delta\phi$ angles in Fig. 4(b). Why are the tails of the distribution so significant? Are there systematic positions on the fiber where these extreme $\Delta\phi$ angles typically occur? Are they from non-specific binding events? Are these measurements correlated with other experimental factors, like PSF overlap or non-uniform background?
 - f. Could ellipticity in the laser polarization state cause some of the asymmetries observed in Fig. 5? Please characterize the ellipticity of the polarization state of the laser with and without the QWP in the sample plane.
 - g. Please report the estimation precision of g_2 for the labeled λ -DNA similarly to the other parameters mentioned in lines 303-305.
 - h. Please quantify the theoretical precision of g_2 estimates as a function of azimuthal angle, polar angle, diffusion coefficient, and axial position, as reported for the other measurement parameters in Supplementary Fig. 1.
 - i. Since the authors use a ROI of 31×31 pixels for ATTO 565 measurements in Fig. 2, I recommend using the same ROI for the quantification of axial precision shown in Supplementary Fig. 1. This will enable the authors to quantify the z estimation precision for $|z| > 300$ nm.
 - j. Please quantify the thickness of the PMMA layers used in this work. In particular, for the data shown in Supplementary Fig. 8, can the authors propose a mechanism for why direct spin coating of Atto molecules produces out of plane polar angles, while immersing them in a PMMA film does not?
3. The simplified orientation model used in this work is presented as complete without referencing other more fully featured models. In particular, Refs. [28,46], Backer (cited below), and Chandler (cited below) fully characterize molecular orientation using 6 basis functions and orientation parameters. These could be the so-called second moments of the transition dipole vector or the coefficients of various spherical harmonics.
- a. Please mention the simplified nature of the model used in this work when it is introduced in line 113 and line 400 of the main text and in Supplementary Note 1.
 - a. Please cite work on more fully featured models:
 1. Backer, A. S. and Moerner, W. E. Determining the Rotational Mobility of a Single Molecule from a Single Image: A Numerical Study. *Optics Express*. **23**, 4255 (2015). doi:10.1364/OE.23.004255
 2. Chandler, T., Shroff, H., Oldenbourg, R., and Rivière, P. La. Spatio-Angular

Fluorescence Microscopy I Basic Theory. *Journal of the Optical Society of America A*. **36**, 1334 (2019). doi:10.1364/josaa.36.001334

3. Chandler, T., Shroff, H., Oldenbourg, R., and Rivière, P. La. Spatio-Angular Fluorescence Microscopy III Constrained Angular Diffusion, Polarized Excitation, and High-NA Imaging. *Journal of the Optical Society of America A*. **37**, 1465 (2020). doi:10.1364/JOSAA.389217

4. There are several issues with missing or mischaracterized references to existing literature that should be corrected.
 - a. Please add a comment and citation to line 60: Localization bias may be removed by using the standard polarized PSF and adjusting the fitting algorithm. Doing so only modestly reduces SNR because of the polarization splitting but does not reduce the density of emitters per frame. See:
Nevskiy, O., Tsukanov, R., Gregor, I., Karedla, N., and Enderlein, J. Fluorescence Polarization Filtering for Accurate Single Molecule Localization. *APL Photonics*. **5**, 061302 (2020). doi:10.1063/5.0009904
 - b. Ref. [31] uses a polarized version of the standard PSF, as in Nevskiy et al. above. Therefore, this technique does not split photons over a large area nor does it limit the density of emitters per frame. In fact, the SNR of these techniques could be superior to the Vortex PSF presented here. Discussion to this effect should be added in line 78 and in line 363.
 - c. Ref. [11] should be added to line 70; this work also used in-focus single-molecule images to measure dipole orientation.
5. More details are needed regarding image analysis:
 - a. How are localizations filtered before being presented in the various analyses in the paper? For example, how are images of overlapping molecules identified and removed?
 - b. How is background fluorescence estimated and fed to the MLE algorithm? Is any spatial or temporal smoothing applied?

Minor comments

1. Please correct the following typographical errors:
 - a. Line 20-21: "Corroborating..." begins an incomplete sentence without a proper subject. Perhaps beginning with "These data corroborate..." is better?
 - b. Line 164: Add a colon ":" after "follows."
 - c. Fig. 2 caption: Please rewrite " $4 - 50 \times 10^3$ " as " $4 \times 10^3 - 50 \times 10^3$ " for clarity.
 - d. There are several instances of hyphenated "single-molecule" that should have spaces instead ("single molecule"). Examples are on line 222 and the Fig. 3 caption.

- e. Line 255: “but the other parameters not that much” is awkward and not a complete sentence. Please rephrase.
 - f. Various figures: correct spelling of “occurance” to “occurrence”.
 - g. Line 432: “aberrations modes” should be “aberration modes”.
 - h. Lines 539-540: “Where we use...” is an incomplete sentence. Please rephrase.
 - i. Supplementary Note 1, line 43: Fix equation running into margin.
 - j. Supplementary Note 1, line 64: “linear” should be “linearly”.
 - k. Supplementary Fig. 5 caption: “Indicating a high precision...” is an incomplete sentence. Please rephrase.
 - l. Supplementary Fig. 6 caption: “FWHM35 nm” is missing an “=” sign.
2. Line 24: “Barely larger” is misleading, since the Vortex PSF is “4-6 times” larger than the standard PSF (line 99). I suggest using “modestly larger” instead.
 3. Lines 114-115: The orientation model should be defined more carefully when it is first introduced. Please define the range of the parameters θ , ϕ , g_2 . Please also state which values of g_2 correspond to free vs. fixed emitters.
 4. Please all panels within Fig. 1 (b,c) on the same color scale. Currently, it is very difficult to compare how distinguishable various features are to one another as a function of PSF and orientation.

In addition, one may argue that the asymmetry of the standard PSF is identifiable by eye (line 130-131). “Difficult to identify” may be better language.

5. Please report the value of g_2 used for the images and characterizations in Fig. 1(b,c,e,f) in the caption of Fig. 1.
6. The characterization of the Vortex PSF is overly simplistic in line 140. The Vortex PSF contains both a central spot and a surrounding weak ring that should be mentioned here.
7. Please define pixel size (65 nm I believe) when pixel regions of interest are first mentioned in line 189.
8. Please define the meaning of $\pm\sigma$ in the Figure 2 caption. Is this one standard deviation?
9. Please add additional quantifications of molecule brightnesses:
 - a. Please add colorbars to Fig. 2(a,c,d) shows image brightnesses in photons/pixel.
 - b. Please add information on the average number of photons detected from the molecule in Fig. 2(a).

- c. Please provide the average and std. dev. of the number of photons detected needed to achieve the precisions reported in lines 231-232.
 - d. Please report the average and std. dev. of the number of photons detected for molecules 1-3 in the caption of Fig. 3.
 - e. Please add a colorbar to Fig. 3(a) showing image brightnesses in photons/pixel.
 - f. Please report the average and std. dev. of the number of photons detected for the precisions stated in lines 303-305.
10. Please report the use of total internal reflection (TIR) illumination in line 310. This aspect is critical to understanding this experiment and should not be relegated to the Methods section.
 11. Both Section 4.1 and Supplementary Note 1 refer to $H(r, \Omega)$, which includes background fluorescence contributions, as a PSF model. However, I believe "PSF" is commonly defined as the response of a microscope to a single point/dipole emitter. Since background must necessarily come from other objects in the sample, it is not correct to refer to $H(r, \Omega)$ as a PSF. It would be better to say "forward model of the imaging system" or simply "image formation model."
 12. Line 444: Please clarify what is meant by "fraction of signal photons captured within the ROI..." Does this statement imply that ~55% of the photons emitted by the molecule are simply not collected by the objective lens? Or, does it imply that ~55% of the photons fall outside of a 15×15 pixel region surrounding the image's center of mass? If it is the latter, then this number seems artificially large. Please explain where this calculation comes from, and comment on why the photons are spread so broadly across the detector.
 13. Line 592: What are the units for the "constant of around 10"? I assume photons?
 14. Supplementary Note 1, line 49: Please define the domain of j . I assume $j=x,y,z$?
 15. Supplementary material: I recommend against using i as an index of summation in equation (4), since it is used as an imaginary number of subsequent equations.
 16. For clarity, please define exactly the aberration function $W(\rho)$ when it is first introduced in line 57 of Supplementary Note 1. I assume it is equal to the sum $K(\rho) + \sum A_n^m(x, y) Z_n^m(\rho)$?
 17. Please adjust the vertical axes of the bias plots in Supplementary Fig. 2 so that these curves are more readable. In particular, the range of g_2 is only -1 to 1, so it is very difficult to read its bias compared to other parameters in degrees in panel (e). The same applies for the intensity bias $\Delta I/I$. Perhaps normalizing the bias by the CRB will make these plots more meaningful.

18. For clarity, please replace “unknown” with “unknown (inaccurately calibrated)” in the caption of Supplementary Fig. 3.
19. Please report the precise $A_n^m(x, y)$ coefficients used for the aberration simulation study in Supplementary Fig. 3.
20. Please comment on the low quality Zernike fittings shown in Supplementary Fig. 4 (h,k,l,n). Why are the reported R^2 values acceptable?

Reviewer #3 (Remarks to the Author):

In "Simultaneous orientation and 3D localization microscopy with a Vortex point spread function," Hulleman et al. demonstrate a quantitative imaging method for determining the 3D positions, orientations, and rotational mobilities of single fluorescent molecules. This method can be readily combined with single-molecule super-resolution microscopy experiments to provide complementary biophysical information about the alignment and ordering of the fluorescent probes, which is not usually detected in conventional experiments. The key ingredient of this technique is a vortex phase mask placed at the Fourier (pupil) plane in the emission pathway of a standard fluorescence microscope. This optic modulates the PSF of the imaging system to yield single-molecule images in which orientation and 3D position related effects are more readily apparent, and amenable to image-fitting techniques. The authors convincingly demonstrate their approach using proof-of-concept experiments conducted on single dye molecules in polymers, and intercalating dyes bound to in-vitro Lambda-DNA (intercalators bind to the DNA by sliding between adjacent base pairs, becoming partially rotationally immobilized). The overall precision of the method is benchmarked using Cramer-Rao lower bound calculations. Furthermore, a thorough characterization of field-dependent optical aberrations is performed, and details regarding the phase-mask alignment procedure are provided. These aspects will greatly assist other experimentalists attempting to reproduce results, and use the method for their own work. It is also worth mentioning that the Vortex PSF can be implemented using off-the-shelf components (more traditionally used for STED microscopy). This will potentially facilitate the widespread adoption of the method.

Unfortunately, it is my responsibility to point out that many similar methods have very recently been published (Refs. 31 and 32 in the submitted manuscript, as well as Lu et al. doi.org/10.1002/anie.202011444). Those papers, among others, have already convincingly demonstrated the core capabilities of 3D orientation/position imaging and precise rotational mobility characterization in a range of samples. I am concerned that this manuscript may be interpreted as an incremental improvement, if the scope is restricted to methods-development and proof-of-concept. However, due to the otherwise extremely high technical quality and promise of the method, I would strongly advocate that the authors be given the opportunity to revise and further extend their work for publication in Nature Communications. This could involve in-depth analysis of their current dataset and additional experimental measurements. It seems like there is a lot more that could be explored in the intercalated Lambda-DNA example. I have accordingly provided a number of suggestions below for directions that I hope the authors may think worthwhile to pursue. Alternatively, an imaging experiment using rotationally fixed dyes in live cells could also be very compelling and serve to better showcase the advantage of combined 3D orientation + 3D position determination (see for example Mehta et al. doi.org/10.1073/pnas.1607674113).

In the authors' Lambda-DNA experiments, they report a difference in $\Delta\phi$ (azimuthal probe orientation w.r.t. DNA-axis) as a function of different laser excitation polarizations. Notably, it is found that by exciting the sample with significantly z-polarized light, the detected intercalators preferentially assume an orientation of $\Delta\phi \sim 65$ deg. (As opposed to 87 degrees when only in-plane illumination is used.) This result is surprising, because it demonstrates there is a significant population of dyes that are not perpendicular to the DNA-axis — which would be highly non-intuitive and add much-needed insight to previous measurements. Furthermore, this result is very interesting because the dyes orient at $\Delta\phi \sim 65$ degrees, but not 115 degrees! As the authors point out, this may be due to the helicity of the DNA structure.

—The asymmetrical preference for $\Delta\phi \sim 65$ over 115 deg. (Fig 5c) is quite remarkable, but this claim really requires more evidence and verification. I would recommend taking data on more DNA aligned along different orientations w.r.t. to the experimental system to try to gauge how reproducible this result actually is. Specifically, if this effect is related to the helicity of the DNA, for some pieces of DNA, one would expect to find a $\Delta\phi$ distribution centered at 115 deg. INSTEAD of 65 deg. Furthermore, if the authors could figure out a way to determine the corresponding AT-rich and GC-rich regions of a given lambda-DNA, this would provide them the means to correlate the intercalator-tilting direction with the directionality of the helix axis of the DNA. Another possibility that the authors should consider is that the observed $\Delta\phi$ is caused

by how the DNA was combed onto the coverslip (I.e. the direction of the receding fluid), and is not related to the directionality of the DNA helix-axis. Helicity effects could be ruled out (or dramatically confirmed) by finding a piece of DNA with an abrupt ~ 180 degree curve, and ensuring that at both sides of the curve $\Delta\phi$ is biased in the same direction, despite the fact that the local helix axis would be running in opposite directions, relative to the experimental reference frame. The other possibility is that the observed tilt could be due to S-DNA formation. This is probably unlikely because the DNA is only slightly extended beyond its contour length, and intercalators will bind to S-DNA-rich regions with much lower affinity. However, I think there are combing protocols that can be used to produce more highly overstretched DNA that could enable a more rigorous study.

—The use of excitation polarization to identify the population of dyes with $\Delta\phi$ close to ~ 65 deg. is a good idea, but I think this could be extended a bit further. Instead of alternating between horizontal and circularly polarized light, it seems like it would be useful to toggle between pure polarization states (purely in-plane and z-polarized states at the sample). This would allow different populations of dyes to be more cleanly isolated. I bring this up because I think some more investigation could be done about the bias in polar (θ) orientations that the authors observe when a vertical (in the plane of the coverslip) excitation polarization is used. While the authors state in words (line 313) that in-plane excitation leads to θ close to 90 deg., their data (Fig. 5b) seems to suggest that θ has a strong out-of-plane orientation of close to 135 deg. (I am assuming that the vertical axis of the 2D histogram in 5b is polar angle, as in 5c, but the panel is not clearly labeled). When using a linear, in-plane excitation polarization, I wonder how θ changes as a function of the orientation of the DNA-axis? For example, would the detected θ orientation histogram change if the DNA is oriented at 0 versus 45 deg. In the experimental reference frame?

—The use of a symmetric rotational mobility parameter (g_2) is a significant improvement over earlier methods that assume fixed dipole emitters, however this may not be the most appropriate rotational-mobility model for the lambda-DNA sample. In this case, I would expect the intercalated dyes to potentially rotate more about the DNA-axis, while remaining roughly parallel to their neighboring base pairs. From the experimental data that the authors have already collected, they could most likely investigate whether there is any asymmetrical wobble evident in the intercalator fluorescent images. This would require slightly more parameters in their image-formation model, but should be entirely feasible given the sophistication of their current analysis efforts.

— A more general point the authors may wish to look into would be to contrast their results using a bis-intercalating dye such as YOYO-1. YOYO-1 has two chromophores, and it is generally assumed that each can be modeled as an independent dipole source. However to my knowledge, a precise single-molecule characterization has not yet been done. Using their image-analysis framework, I think it would be extremely compelling if the authors could determine the individual orientations of the two YOYO-1 chromophores within a single dye, as well as their relative tilt angles with respect to the DNA-axis. This type of characterization could potentially help to identify probes that report more information regarding local deformation in the DNA (e.g. changes in the intra-molecular orientation between two intercalated chromophores would be useful for identifying acute changes in DNA structure over ~ 4 base-pair intervals).

Signed: ADAM BACKER

We thank the reviewers for their careful consideration of our manuscript and the many valuable suggestions, at some points thinking broader than the direct scope of the paper. In the point-by-point response we have indicated that we have investigated the different mentioned possibilities to widen the scope and applicability beyond a proof-of-principle, as e.g. suggested by reviewer 3. Eventually, we decided, however, for another biological application than suggested namely the imaging of supercoiled DNA, or plectonemes. In order to do so, we sought collaboration with Cees Dekker and Eugene Kim, who are now also co-author. The key results of this additional application are shown in the new Figure 6.

Apart from addressing the questions and requests from the reviewers we also made two minor improvements to our data analysis, improving results a bit.

- 1. We found that the lambda-DNA sample provides a way to make a coarse-to-fine drift correction. Initially we only did a coarse drift correction using the Picasso software. After studying the resulting image data in some detail we found correlations between the deviations in localizations from the spline fitted DNA strands as a function of time between near parallel straight strands at different positions in the FOV. We then used these correlations to make a fine drift correction, reducing the estimated residual drift between 5-10 nm to numbers below 5 nm. This is detailed in section 4.7 on data analysis and in Supplementary Figure 19.*
- 2. We also looked at the lambda-DNA images filtered for polar angles below and above 90 deg, and found that there could be biases in the range 5-10 nm. We checked the sensitivity to all optical parameters and found that the assumed refractive indices for medium, cover slip and immersion oil can have such an impact on the accuracy if they vary on the order of $\sim 10^{-2}$. We then tuned one of the refractive indices (for the coverslip) to minimize the bias. This is further described in section 2.5 on the lambda-DNA experiments, starting line 347, and in Supplementary Figure 7. The claims on localization accuracy we make now have an added disclaimer on assumed refractive indices and the associated spherical aberration control. In the discussion we state "Using the Vortex PSF we achieve accurate parameter estimation, avoiding the large position biases of several 10s of nm commonly seen with fixed dipole emitters in localization microscopy. Instead we reduce these biases to below the localization precision by matching experimental parameters like the refractive index, if one uses the nominal values for these quantities we find only a small bias of up to several nm."*

The result of these improvements is that the measured FWHM of the distribution of localizations w.r.t. the spline fitted DNA lines is about 18 nm, which can be attributed largely to the localization precision of around 5.4 nm (sigma to FWHM is a factor 2.4).

Detailed response to the reviewers:

Reviewer #1 (Remarks to the Author):

In their manuscript, Rieger and coworkers present a very cool application of the vortex phase plate in single-molecule localisation microscopy: it allows engineering PSFs such that not only location in X, Y, and Z can be determined, but that also the orientation for the molecule (θ , ϕ , and g_2) can be determined, with high accuracy and up to the statistical limit. Overall, the explanation, the physical foundation and the demonstrations are very sound and clear, also for the broader Nature Comm audience. I think this is a very interesting new tool for the single-molecule community, it appears quite easy to implement in many setups and the authors provide detailed information for this including fitting routines. I wholeheartedly recommend publication in Nature Comm.

Thank you.

I would like to make a few comments that might help the authors to make their manuscript even better.

Science:

- The authors show that a SBR of at least 200 is required. This is pretty high, but that is of course not surprising, 6 parameters should be extracted from a single image. Could the authors discuss (in the end?) this a bit more, describing what this would practically mean and to what limitations this leads (e.g. time resolution / bleaching etc. in the DNA intercalator experiments).

The requirement of an SBR of 200 is determined by fixing the background to 10 photons per pixel (typical for SMLM) and sweeping the signal photon count. Therefore this requirement is mainly limited by the signal photon counts. For a higher signal photon count, for example 4000 photons a worse SBR can be tolerated and the estimator still converges properly with an SBR of 40. For the DNA intercalator experiments this is not a concern as the SBR is typically 450. We now mention in the discussion "The SBR requirement of at least 200 (2000 signal photons and 10 background photons) is met in typical SMLM experiments, and allows relatively short exposures of ~ 30 ms to be used." We also would like to mention that we fit 8 parameters, not 6, namely coordinates x_0 , y_0 , z_0 , photon counts N , b , and orientational parameters θ , ϕ , and g_2 .

- I think it is / could be a bit confusing that the authors use the same θ / ϕ for the DNA experiments and for the other experiments (and theory), without explicitly mentioning it. For the DNA, the DNA axis is taken as the central axis, for the other experiments it is the optical axis (which is perpendicular to the DNA axis). I mean, this is fair enough, but maybe the reader would be helped to explicitly remark this. A further question is what this "coordinate transformation" does to the errors in the angles that are determined.

The concept of the relative angle $\Delta\phi$ was indeed not properly defined. We have now included a definition of this in the section 2.5: "The local orientation of the DNA strand can be determined by fitting a spline curve to the localizations on the strand. Thereby the angle difference ($\Delta\phi$) between the fluorophore and DNA axis can be estimated, i.e. $\Delta\phi$ is the azimuthal angle in a coordinate frame where the DNA strand is locally pointing in the x-direction." Additionally a diagram of the definition of $\Delta\phi$ is included in figure 4a. We hope this solves the misunderstanding that we use a coordinate system with the DNA axis as z-axis instead of the optical axis. This is not what we do, as such the polar angle thus remains unchanged and still is the angle with the optical axis. The (local) coordinate transformation (rotation around z such that locally the strand points along x) does not induce any additional errors as the spline curve fit to the

localization data is more precise than the drift or the localization estimation and the DNA strands appear to have low curvature.

- The lambda DNA experiments. The authors hypothesize that the heterogeneity they observe might be to a fraction of SDNA in their combed DNA molecules. This could indeed be the case. It would be very, very cool to test this in optical tweezers / fluorescence experiments, but that is out of the scope of this paper (but I would recommend the authors to do this in a follow up). Maybe the authors could consider (in the current manuscript) other explanations / effect: could the SDNA be dynamically / locally formed and disappear (and thus be less location specific)? Might there be other effects due to torsionally constraining the DNA and inducing supercoiling due to intercalator binding?

With optical tweezers very interesting experiments could indeed be devised where the DNA is stretched and then attached to a raised surface and imaged with the Vortex PSF. Unfortunately directly imaging Sytox Orange on DNA in an optical trap with the Vortex PSF might not be productive due to the twirl of the DNA around its own axis. Considering the attachment of the DNA to the coverslip it seems unlikely to change state over time without applying additional forces, for example by adding or removing large concentrations of intercalators.

We have now also included a sample that is saturated with Sytox Orange in advance to induce torsion in section 2.6. This results in super-coiled strands and the formation of plectonemes. These super-coiled strands show a different preferential dipole binding orientation than the lambda DNA. Additionally a periodicity of localizations with certain dipole orientations was found.

Text:

- Abstract: "Developed an engineered..." Don't the authors use a widely used (for STED) and commercially available device in another way than before, opening up a great new application? To me these beginning words do not really describe what the authors did.

We have adjusted the entire abstract to better describe what we did. In particular the first two sentences have been changed to "Utilization of a modified Point Spread Function (PSF) enables the simultaneous estimation of dipole orientation, 3D position, and degree of rotational constraint of single molecule emitters from a single 2D image plane. We use an affordable and commonly available phase plate, normally used for STED microscopy in the excitation light path, to alter the PSF in the emission light path." The use of the same phase plate for STED and MINIFLUX is mentioned in the methods sections, however as this is important to understand the function of the phase plate we have also included this in the introduction of the concept: "This phase plate is commonly used in STED microscopy to alter the excitation PSF, here we use it to engineer the emission PSF instead."

- Line 61: "Instead of only avoiding..." I did not understand this sentence and how it connects to sentences before and after.

This sentence connected the previous paragraph about avoiding localization bias and tried to connect it to the next paragraph about also trying to determine the orientation of the molecules. This sentence has now been changed to "A polarized standard PSF [20,21] can not only be used to avoid localization bias but also to identify the in-plane orientation of fixed emitters.." We feel that the transition now also makes more sense with additional references suggested by reviewer 2.

Reviewer #2 (Remarks to the Author):

1

Review of Nature Communications manuscript NCOMMS-20-43805

“Simultaneous orientation and 3D localization microscopy with a Vortex point spread function”

In this work, Hulleman, et al., develop an engineered point spread function (PSF), called the Vortex PSF, to simultaneously measure the dipole orientation, 3D position, and degree of rotational constraint of single-molecule emitters. A vortex phase plate inserted the microscope's back focal plane (BFP) changes the image of a single molecule into a donut-like shape. The vortex PSF has superior performance for estimating the 3D position and orientation of single molecules compared to the standard PSF, as quantified by the Cramér-Rao bound (CRB). It is also able to discern re-orientation dynamics of single Atto 565 molecules spin-coated on glass. Finally, the authors demonstrate imaging the positions and orientations of DNA-intercalating dyes on λ -DNA. They find an interesting correlation between the polar angle of the dye versus its binding rigidity to the DNA.

Overall, the authors present an intriguing new method for adding dye orientation information to single-molecule localization microscopy (SMLM). The technique is a clever reuse of an existing PSF, and the authors carefully characterize their technology both in theory and in experiments.

Thank you.

However, the long-term significance of the method, as demonstrated, is still unclear. Additional comparative evidence to existing states of the art should be added before a final decision can be reached. Detailed recommendations for improvements are below.

Major comments

1. The suitability of the vortex PSF for various SMLM applications is a little unclear from this work. In particular, the performance of the vortex PSF (estimation precision, predicted signal-to-noise ratio (SNR), etc.) is quantitatively compared to only the standard PSF. The lack of other comparisons is problematic because, as the authors state, the standard PSF is only suited for estimating the orientations of very bright fluorophores or fluorophores with certain Orientations.

The numbers supplied along with Supplementary Figures 1, 2 and 3 can be used to compare the Vortex PSF to other methods. To do this explicitly we have now included a table (Supplement, Table 1) comparing the Vortex PSF's theoretical precision limit (CRLB) to other methods using any parameters mentioned by the other methods. The advantages and disadvantages of the Vortex PSF as evident from this table are summarized in the discussion (section 3) which has been restructured. In particular the 2nd paragraph: “To put the precision of the Vortex PSF into perspective we have made a comparison to state-of-the-art PSF-engineering methods for orientation estimation in Supplementary Table~1. Compared to the Tri-spot PSF[30], the Vortex PSF has a similar performance in the azimuthal angle, however the polar precision is significantly better. Methods focused on polarization splitting, for example a polarized PSF[21] or CHIDO[33], benefit from a better azimuthal precision as distribution of photons between the two polarization channels reveals more information about the polarization of light and consequently the orientation of the molecular dipole moment. This also appears to yield a slightly better precision in the wobble cone semi-angle α for methods with polarization splitting. Overall there is not a lot of variation in the lateral localization precision as all but the Tri-spot PSF are fairly compact PSFs.

It is important to realize that currently most methods except the Vortex PSF and CHIDO cannot estimate the full 3D position along with both orientation angles and the rotational constraint. Comparing the only two methods that could estimate all these parameters simultaneously, CHIDO has the advantage that the polarization splitting improves the azimuthal and axial precision. The estimator used in CHIDO, however, only gets to within 200%-500% of the theoretical CRLB limit on simulated PSFs. On experimental data this is further reduced to 500%-600% (uncertainty_{xy}=13 nm CRLB_{xy}=2.3 nm, uncertainty_z=50 nm CRLB_z=7.5 nm, uncertainty_{theta}=5 degrees CRLB_{theta}=0.8 degrees, uncertainty_{solidangle}= 0.9 CRLB_{solidangle}=0.13). This could be caused by aberrations or by difficulties in characterizing the birefringence of their stress-engineered optical element. The estimation of parameters with the Vortex PSF reaches the CRLB in simulation over a wide range of signal to background ratios as shown in Supplementary Fig. 2. On experimental data we have shown that the Vortex PSF achieves a precision up to 30% above the CRLB on fixed molecules (section 2.3) and close to the CRLB on intercalators attached to DNA (Supplementary Figure 10). Finally, the Vortex PSF has the virtue of simplicity in optical setup compared to the Tri-spot PSF and CHIDO, which both use a custom phase-mask and polarization splitting. Summarizing, the polar precision is better for the vortex PSF compared to other compact PSF designs, whereas the azimuthal precision is slightly worse compared to more experimentally complex polarization splitting methods. The Vortex PSF could be expanded to utilize polarization splitting and potentially improve both the azimuthal and axial precision. This would come at the cost of the simplicity of the optical setup.”

a. Please make the comparison in Fig. 1(e,f) more robust by computing the orientation estimation precision (CRB) of some combination of the following:

The goal of fig. 1 (e,f) is to highlight the difference between the Vortex PSF and the standard non-engineered PSF and why inverting the phase relationship helps. To avoid making this figure more complicated we have not included the comparison here. In order to answer the reviewer's requests for further comparison we have instead included a comparison table (Supplementary Table 1) that compares the CRLB of other methods to ours, added Supplementary Figure 3 with a detailed comparison to defocused orientation fitting and discuss the advantages and disadvantages of our method in section 2.2 and the discussion (section 3).

i. A modestly defocused standard PSF, which is very easy to implement. There should be some intermediate defocus values that enable robust localization and orientation measurement, similar to Refs. [20-24]. Please comment on if the Vortex PSF is uniformly better than all possible defocused standard PSFs, or if the standard PSF has superior performance for some set of defocus values and Orientations.

We have added Supplementary Fig. 3 that compares the Vortex PSF and defocused orientation fitting over a 3 micron range. In the main text (section 2.2 starting line 216) we describe what can be seen from this figure. In short, the Vortex is better around focus $abs(z) < 300$ nm. In practice this means that the Vortex PSF can be used over a larger z-range spanning both the defocused range $300 < z < 1000$ nm and the region around focus $-300 < z < 300$ nm.

ii. A polarized standard PSF (as used in Ref. [46]), which only needs a polarizing beamsplitter or a dual-channel polarization module like the Opto-Split from Cairn. Ref. [46] showed that both in-plane orientation and rotational mobility g_2 could be robustly measured for lower photon counts than reported here. Please comment on the strengths and weaknesses of the Vortex PSF compared to this technique.

Ref. [46] (<https://doi.org/10.1103/PhysRevLett.122.198301>) seems to deal with the theoretical precision in estimating the 2D and 3D rotational diffusion of various techniques but it does not appear to include the standard polarized PSF. Some of the parameters in this paper can be estimated at lower photon counts because excitation polarization modulation is utilized. In this case a priori information (excitation polarization direction) is available that improves the estimation. The other methods illustrated in Ref. [46] supplementary figure 6-8 have similar rotational diffusion precision compared to the Vortex PSF (σ^2 precision is now included in our supplementary figure 1).

Compared to a polarized standard PSF like in Ref. [21] (<https://doi.org/10.1364/OPTICA.388157>) the required photon counts or SBR is similar. On simulated data the authors of Ref. [21] claim to need 300 signal photons and 2 background photons per pixel (SBR = 150). With the same settings the Vortex PSF converges to within 15% of the CRLB for all parameters. The requirement of SBR = 200 as mentioned in the main manuscript is calculated with a typical SMLM background of 10 photons per pixel and therefore appears to need higher photon counts. We have included a comparison to Ref. [21] in Supplementary Table 1 and discussed the strengths and weaknesses of the Vortex PSF compared to it in the discussion (section 3, line 499-535).

iii. CHIDO (as used in Ref. [32]). From Ref. [32], it seems that CHIDO is much more precise than the Vortex PSF for the same number of photons detected. Please comment on the strengths and weaknesses of the Vortex PSF compared to this Technique.

A comparison between the Vortex PSF and CHIDO has been made in Supplementary Table 1 and the differences are discussed in the manuscript (Section 3 discussion, 2nd paragraph). It appears that the precision of CHIDO is about twice as good in the azimuthal angle and z position. This is in a large part due to the polarization splitting used in CHIDO. If the complexity of the setup is of no concern the Vortex PSF could in the future be expanded to also utilize polarization splitting to improve in particular the azimuthal and z precision. A theoretical precision that is twice as good but in practice not attained is however not so useful. The reported experimental precision with CHIDO is 500%-600% higher than the theoretical limit (CRLB) (uncertainty_{xy}=13 nm CRLB_{xy}=2.3 nm, uncertainty_z=50 nm CRLB_z=7.5 nm, uncertainty_{theta}=5 degrees CRLB_{theta}=0.8 degrees, uncertainty_{solidangle}= 0.9 CRLB_{solidangle}=0.13). With the Vortex PSF the experimental precision can be compared with Supplementary Figure 10 where we analyse repeated localizations during a single on-event, and with the proof-of-principle experiments of section 2.3 (Figure 2) where we assess angular precision over the different z-slices. Based on this data we find that the precision is within 30% above the theoretical limit (CRLB). Therefore in practice the precision obtained with the Vortex PSF is about two times better than the current implementation of CHIDO.

iv. The analyses in the following preprint seem to be relevant. Is the Vortex PSF superior to any of the methods mentioned? Single-molecule orientation localization microscopy II: a performance comparison Oumeng Zhang, Matthew D. Lew arXiv:2010.04064

The thorough performance comparison presented above quantifies the orientation precision with one angle. As the methods compared vary in azimuthal precision and polar precision we opted to create a comparison (Supplementary Table 1) to the original manuscripts mentioned in the mentioned arxiv preprint instead. The performance of the Vortex PSF is typically better in the polar angle compared to the other methods.

b. Lines 362-363: The statement “the CRLB for the Vortex PSF is good for all possible emitter orientations” does not currently have adequate quantitative evidence to

support it. The comparisons requested above are very important to include, and this statement should be revised to reflect the new observations.

We have changed this statement slightly but stand by our original intention to say that the precision with the Vortex PSF is good independent of the emitter orientation, as such the CRLB does not diverge for any particular orientation. The statement has been changed to: "... and the CRLB for the Vortex PSF is relatively uniform for all possible emitter orientations." We have included the comparisons suggested above and reflected on the observations in the manuscript. In summary, the polar precision is better with the Vortex PSF and the azimuthal precision is slightly worse due to the lack of polarization splitting, which we now clearly state in the discussion section 3.

2. Several of the experimental demonstrations need additional quantification/analysis to support the conclusions of this work.

a. Please report the mean and standard deviation of g_2 estimates from the example molecule in Fig. 2 and the 21 molecules in Supplementary Fig. 5. Are these measurements consistent with molecules confined within PMMA? If there are discrepancies, please discuss possible causes for them.

We have added the g_2 estimates of this particular molecule to the caption of Fig 2. The average g_2 and standard deviation of the 21 other molecules have been added in the text and compared to literature values, which all show similar degrees of rotational constraint: "The mean rotational constraint is $g_2 = 0.86$, with a standard deviation of 0.03. This corresponds to an average wobble cone semi-angle of 25.3 deg which is similar to the previously found rotational constraint of single molecules in PMMA ~ 0.85 [38] ($\alpha = 27.8$ deg) and ~ 0.8 [30] ($\alpha = 30.7$ deg) "

b. Please include raw images of molecule 2 and molecule 3 before and after their orientation "jumps" in Fig. 3 (i.e., frame 9 for molecule 2 and frame 18 for molecule 3). Please add an annotation to the various trajectories in Fig. 3(c) showing the position of the 3 molecules when the orientation "jump" occurred. Is there any correlation between the orientation and position trajectories of each molecule?

Figure 3 has been adapted to include raw images of all 3 molecules before and after one of their transitions. The annotations have been added in figure 3(c). An additional quantification of the position shift has been given in the text (standard deviation of localizations and the CRLB): "There does appear to be a marginal position shift as the molecules re-orient, however the average standard deviation of the localizations $xy = 2.7$ nm and the $CRLB_{xy} = 1.7$ nm show that the residual shift is small compared to the theoretical precision limit." This shows that the minute position shift is very close to the theoretical localization precision.

c. Please clarify the meaning of and rationale for reporting Δz in Fig. 3(c). It seems to me that it would be more useful to report here the absolute z position of each molecule with respect to the focal plane.

This was done to align the trajectories on top of each other. As the coverslip is not perfectly perpendicular to the optical axis there is a variation in the z -position throughout the field of view. The figure is now adapted with the actual z position of each molecule with respect to the focal plane.

d. Fig. 4(e) reports the z positions of labeled λ -DNA without any quantification or analysis. What is the measured tilt angle? Is this degree of tilt expected given the apparatus used in the experiment? Does the sample appear to be perfectly planar, i.e., flat? What is the variation in measured z positions, and how much do they deviate from the CRB? Overall, I find this axial localization demonstration to be weak, and a more compelling demonstration would improve the paper.

The analysis of the tilted lambda-DNA sample has been expanded to include fitted plane surfaces through the localizations and deviations from these surfaces in Supplementary Fig. 9. In the main text we report the retrieved surface tilt difference (0.42 degrees) which is very close to the applied tilt of 0.4 degrees. The planar surfaces fit the data well and the variation as shown in Supplementary Fig. 9 is typically 40 nm RMSE or 64 nm FWHM which is to be expected from a mean z CRLB of 25 nm. These distributions have small tails that we attribute mainly to contributions from very freely rotating dipoles.

e. Please comment on the non-Gaussian distribution of the measured $\Delta\phi$ angles in Fig. 4(b). Why are the tails of the distribution so significant? Are there systematic positions on the fiber where these extreme $\Delta\phi$ angles typically occur? Are they from non-specific binding events? Are these measurements correlated with other experimental factors, like PSF overlap or non-uniform background?

We analyzed the distribution in some more detail. It appears that the distribution can be well fitted with a Gaussian plus a constant (Rsquare=0.964).

The tails do not appear significant, we interpret that they are primarily from non-specific binding occurring throughout the analyzed field of view. Only a relatively small fraction of the dipoles oriented perpendicular to the average orientation are located on the DNA strand. These occur in 2 clusters where presumably the spline fit doesn't exactly match the actual DNA location.

If there is significant PSF overlap or non-uniform background then the quality of fit will be low, as measured by a high Chi-Square value. In our algorithm these localization events are removed (see section 4.7 on data analysis), values outside the range $0.75 < \chi^2 < 3$ are rejected.

f. Could ellipticity in the laser polarization state cause some of the asymmetries observed in Fig. 5? Please characterize the ellipticity of the polarization state of the laser with and without the QWP in the sample plane.

To quantify the polarization state of the incident laser beam we use a linear polarizer in a coarse rotation stage and a power meter. The polarization without the QWP is perpendicular to the plane of incidence (s-polarized, $\phi=90$ degrees) with an extinction ratio of 240:1. Deviations from the intended linear polarization are therefore expected to be very small, whether in TIRF or in EPI illumination. With the QWP inserted the extinction ratio is 3:1 with a polarization angle of $\phi=60$ degrees (~30 degrees tilted compared to without the QWP). This corresponds to an approximate orientation of $\theta \sim 15$ and $\phi \sim 90$ (it is more out-of-plane due to the increased intensity of the z component) in TIRF illumination. The in-plane orientation (ϕ) in TIRF conditions can however vary easily if the beam angle is not exactly at the critical angle. The polarization after the QWP is not perfectly circular (1:1) which is probably due to the waveplate in the Nikon TIRF illuminator being misaligned and unfortunately glued in place allowing no adjustability, alternatively the QWP might not be achromatic enough.

If the ellipticity or the polarization state was the cause for any observed asymmetries then the histograms would depend on the local orientation of the DNA strand which is not the case. This was mentioned in the initial manuscript but to illustrate this further we have now added Supplementary Fig. 11 where the distribution is the same for 4 additional DNA strands with some slight deviations around $\theta=90$ as certain orientations are less efficiently excited.

g. Please report the estimation precision of g_2 for the labeled λ -DNA similarly to the other parameters mentioned in lines 303-305.

We have added the estimated g_2 precision for the labelled lambda-DNA in Supplementary Fig. 10 along with the estimated photon count distribution suggested by Reviewer #3. We find that the g_2 precision on the λ -DNA data is 0.06 which is now mentioned in the main text.

h. Please quantify the theoretical precision of g_2 estimates as a function of azimuthal angle, polar angle, diffusion coefficient, and axial position, as reported for the other measurement parameters in Supplementary Fig. 1.

We thank the reviewer for this suggestion, and we have included the third column in Supplementary Fig. 1 that shows the precision of g_2 estimates as a function of the investigated parameters.

i. Since the authors use a ROI of 31×31 pixels for ATTO 565 measurements in Fig. 2, I recommend using the same ROI for the quantification of axial precision shown in Supplementary Fig. 1. This will enable the authors to quantify the z estimation precision for $|z| > 300$ nm.

A ROI of 31x31 can only be used for sparse samples, which was the case for the ATTO 565 sample. A ROI of 15x15 is more applicable for dense single-molecule localization microscopy and therefore we have kept the ROI of 15x15 for Supplementary Fig. 1. We have created an additional Supplementary Figure 3 to evaluate the precision over a more extended axial range with a ROI of 31x31, imaging in either matched conditions or an aqueous medium. The figure compares the Cramér–Rao bound of the estimated parameters for standard PSF and the vortex PSF over a 3 μ m axial range. The phase inversion of the vortex phase plate has a clear advantage around focus, $|z| < 500$ nm, as the large CRLB variation is removed.

j. Please quantify the thickness of the PMMA layers used in this work. In particular, for the data shown in Supplementary Fig. 8, can the authors propose a mechanism for why direct spin coating of Atto molecules produces out of plane polar angles, while immersing them in a PMMA film does not?

In the initial Supplementary Figure 8 (now Supplementary Fig. 12) it is not whether or not the molecules were immersed in PMMA that determines the most occurring orientation. As specified in the caption, in the single-molecule case without PMMA the excitation is in TIRF conditions and therefore primarily excites out-of-plane orientations (polar angles close to 0 and 180 degrees). For the single molecules in PMMA epi-illumination is used, as there is almost no refractive index difference between glass and PMMA making TIRF illumination impossible. This does raise an important question, namely why the orientations in the main Figure 3 are primarily out-of-plane. Our answer is that this is because the in-plane orientations were imaged before this image series (in epi-illumination and showing no reorientation) and bleached in the process. This is now mentioned in the main text: “Initially the region is imaged with epi-illumination which bleaches the in-plane molecules which showed almost no re-orientation.” Unfortunately we do not have access to an ellipsometer to accurately determine the thickness of the PMMA layer.

3. The simplified orientation model used in this work is presented as complete without referencing other more fully featured models. In particular, Refs. [28,46], Backer (cited below), and Chandler (cited below) fully characterize molecular orientation using 6 basis functions and orientation parameters. These could be the so-called second moments of the transition dipole vector or the coefficients of various spherical harmonics.

a. Please mention the simplified nature of the model used in this work when it is introduced in line 113 and line 400 of the main text and in Supplementary Note 1.

a. Please cite work on more fully featured models:

1. Backer, A. S. and Moerner, W. E. Determining the Rotational Mobility of a Single Molecule from a Single Image: A Numerical Study. *Optics Express*. 23, 4255 (2015). doi:10.1364/OE.23.004255

2. Chandler, T., Shroff, H., Oldenbourg, R., and Rivière, P. La. Spatio-Angular Fluorescence Microscopy I Basic Theory. *Journal of the Optical Society of America A*. 36, 1334 (2019). doi:10.1364/josaa.36.001334

3. Chandler, T., Shroff, H., Oldenbourg, R., and Rivière, P. La. Spatio-Angular Fluorescence Microscopy III Constrained Angular Diffusion, Polarized Excitation, and High-NA Imaging. *Journal of the Optical Society of America A*. 37, 1465 (2020). doi:10.1364/JOSAA.389217

These more full featured rotational diffusion models have now been included in the reference list. The appropriateness of our “simplified” rotational diffusion model is extensively discussed in reference [36] (<https://doi.org/10.1364/JOSAA.32.000213>). Combining this rotational diffusion model with a fully vectorial PSF model that accounts for refractive index mismatches and aberrations, accurately matches the experimental data. The main limitation or simplification of our rotational diffusion model is that we assume symmetric rotational diffusion, which depending on how the fluorophore is constrained is not necessarily the case. The first suggested manuscript (<https://doi.org/10.1364/OE.23.004255>) does present an interesting model for asymmetric diffusion which we have explored in an answer to reviewer 3 below. In the fitting model section of the methods we have included the following: “This seems simplistic compared to more complex rotational diffusion models [54, 48, 55, 56]. The appropriateness of the model for rotational diffusion faster than the fluorescence lifetime, however, is demonstrated in ref. [36]. An asymmetric rotational diffusion model would require 2 additional fitting parameters, raising the total amount of parameters to 10 which we expect is not realistic to fit with < 5000 photons.

In the discussion we have included: “The feasibility of estimating the orientational confinement that is not rotationally symmetric around the preferential axis defined by the minimum of the orientational potential well in addition to the other parameters could also be investigated. Going from a uniaxially symmetric to a biaxially symmetric orientational confinement would bring the number of orientational parameters that must be estimated from three to five. Aside from the general degree or rotational constraint and the primary dipole orientation an additional parameter is needed to describe the primary direction of rotational diffusion and another for the degree of asymmetry. A reliable estimation of the then total number of 10 parameters (instead of 8) may require a more complex setup involving e.g. polarization detection in addition to the vortex phase plate.”

4. There are several issues with missing or mischaracterized references to existing literature that should be corrected.

a. Please add a comment and citation to line 60: Localization bias may be removed by using the standard polarized PSF and adjusting the fitting algorithm. Doing so only modestly reduces SNR because of the polarization splitting but does not reduce the density of emitters per frame. See:

Nevskyi, O., Tsukanov, R., Gregor, I., Karedla, N., and Enderlein, J. Fluorescence Polarization Filtering for Accurate Single Molecule Localization. *APL Photonics*. 5, 061302 (2020). doi:10.1063/5.0009904

We have added the following around the original line 60: “Alternatively, the polarization could be split into an x and y component and the y and x position could then be fitted from each respective polarization channel [20]. This also avoids the localization bias but results in an asymmetric localization precision. A polarized standard PSF [20, 21] can not only be used to avoid localization bias but also to identify the in-plane orientation of fixed emitters.” Unfortunately the localization precision with the method from Nevskyi et al. can be very asymmetric depending on the molecule dipole orientation.

b. Ref. [31] uses a polarized version of the standard PSF, as in Nevskyi et al. above. Therefore, this technique does not split photons over a large area nor does it limit the density of emitters per frame. In fact, the SNR of these techniques could be superior to the Vortex PSF presented here. Discussion to this effect should be added in line 78 and in line 363.

The context of ref [31] (<https://doi.org/10.1364/OPTICA.388157>) in line 78 is the fact that the tri-spot PSF splits the light over 6 spots and has therefore a problematic detectability at an SBR where the polarized PSF can still be used (as literally

mentioned in ref [31]). The use of a polarized PSF as used in ref [31] and by Nevskyi et al. has been added in the introduction (original line 78) (see quote in answer above) and discussion (original line 363): “In particular the polar precision is better compared to other compact PSF designs, however the azimuthal precision is slightly worse compared to more experimentally complex polarization splitting methods. The Vortex PSF could be expanded to utilize polarization splitting and potentially improve the azimuthal and axial precision at the cost of simplicity of the optical setup.”

Splitting the polarization can improve the azimuthal precision but the standard polarized PSF cannot be used to determine the polar angle as seen by the poor polar precision shown in Supplementary Figure 11 of ref [31] and by the linear bias in polar angle shown in Supplementary Figure 12 of ref [31].

c. Ref. [11] should be added to line 70; this work also used in-focus single-molecule images to measure dipole orientation.

This is indeed the case and has been added.

5. More details are needed regarding image analysis:

a. How are localizations filtered before being presented in the various analyses in the paper? For example, how are images of overlapping molecules identified and removed?

In section 4.7 on data analysis we now list the filtering that we do: “In the fitting routine potential outliers and poor localizations are filtered out as follows. The number of iterations in the MLE is terminated if the relative difference between successive log-likelihood values in the iteration drops below 10^{-6} or if the maximum number of iterations, which is set equal to 30, is reached. Typically localizations converge within 20 iterations, events are rejected if the maximum number is reached without convergence. Moreover, when the estimated molecule position is more than 3 pixels away from the center pixel in the ROI, it is rejected. Finally, converged localizations are tested with a normalized chi-squared. If a localization has a χ^2 value outside the region $0.75 \leq \chi^2 \leq 3$, it is also rejected. Additional filtering is specified where used.”

b. How is background fluorescence estimated and fed to the MLE algorithm? Is any spatial or temporal smoothing applied?

We do not apply any smoothing to the images. The background is a constant off-set in the imaging model, as indicated in equation (1), and treated as an estimation parameter in the MLE algorithm applied to any of the ROIs containing single molecule spots. Hence it is not fed to the MLE algorithm but estimated in the same iterative process as the other parameters.

Minor comments

1. Please correct the following typographical errors:

a. Line 20-21: “Corroborating...” begins an incomplete sentence without a proper subject. Perhaps beginning with “These data corroborate...” is better?

We thank the reviewer for this suggestion and have rephrased the sentence to: “These data corroborate perpendicular azimuthal angles to the DNA axis for in-plane emitters...”

b. Line 164: Add a colon “:” after “follows.”

This has been added.

- c. Fig. 2 caption: Please rewrite “4 – 50 × 103” as “4 × 103– 50 × 103” for clarity.
This has been corrected.
- d. There are several instances of hyphenated “single-molecule” that should have spaces instead (“single molecule”). Examples are on line 222 and the Fig. 3 caption.
This has been changed.
- e. Line 255: “but the other parameters not that much” is awkward and not a complete sentence. Please rephrase.
We have rephrased the sentence to “The three different molecules that undergo these re-orientation events seem to primarily make jumps in the azimuthal angle. The re-orientations are also observed in the rotational constraint g_2 as it takes a lower value in the transition frames, which indicate a more freely rotating molecule or a superposition of orientations before and after the transition.” which we feel makes the flow of the text more natural.
- f. Various figures: correct spelling of “occurance” to “occurrence”.
This has been corrected in Figure 4 and relevant supplemental figures.
- g. Line 432: “aberrations modes” should be “aberration modes”.
This has been corrected to: “aberration modes”
- h. Lines 539-540: “Where we use...” is an incomplete sentence. Please rephrase.
This sentence was redundant and has been removed.
- i. Supplementary Note 1, line 43: Fix equation running into margin.
This and another instance have been fixed.
- j. Supplementary Note 1, line 64: “linear” should be “linearly”.
This has been adjusted to: “linearly”
- k. Supplementary Fig. 5 caption: “Indicating a high precision...” is an incomplete sentence. Please rephrase.
This has been rephrased to: “These estimates indicate no bias between the Vortex PSF and through-focus orientation estimation”
- l. Supplementary Fig. 6 caption: “FWHM35 nm” is missing an “=” sign.
We thank the reviewer for catching this error and have added the “=” sign
2. Line 24: “Barely larger” is misleading, since the Vortex PSF is “4-6 times” larger than the standard PSF (line 99). I suggest using “modestly larger” instead.
This would indeed be misleading if it was 4-6 times larger but the Vortex PSF is 4-6 times the Rayleigh criterion in width. Where the standard PSF is 2-4 times the Rayleigh criterion in width. Therefore the size difference between the two is only 1.5-2 times depending on the orientation of the emitter. We have adjusted the abstract to: “The Vortex PSF does not require polarization splitting and has a compact PSF size, making it easy to implement and combine with localization microscopy techniques.” and made the size difference more explicit in the introduction: “This is because the single imaged spot with the Vortex PSF has a footprint of only 4-6 times the Rayleigh criterion in size, compared to a standard non-engineered PSF that has a width of 2-4 times the Rayleigh criterion. Therefore, depending on the emitter’s orientation, the Vortex PSF is only 1.5-2 times larger than a standard non-engineered PSF.”

3. Lines 114-115: The orientation model should be defined more carefully when it is first introduced. Please define the range of the parameters θ , ϕ , g_2 . Please also state which values of g_2 correspond to free vs. fixed emitters.

This has been added where the parameters are first introduced: "In the completely freely rotating case $g_2 = 0$ and for a fully constrained molecule $g_2 = 1$. The angle parameters ϕ and θ can take any value, however presented results are mapped to a single hemisphere; ($0 \leq \theta < 180$ & $0 \leq \phi < 180$)."

4. Please all panels within Fig. 1 (b,c) on the same color scale. Currently, it is very difficult to compare how distinguishable various features are to one another as a function of PSF and Orientation.

The panels are now on the same colour scale, with the maximum intensity scaled between the normal PSF and Vortex PSF. Without the normalization for each PSF shape it is harder to discern any azimuthal orientation in the standard PSF. It does illustrate how segmentation with the Vortex PSF is easier as there is less intensity variation as a function of the polar angle.

In addition, one may argue that the asymmetry of the standard PSF is identifiable by eye (line

130-131). "Difficult to identify" may be better language.

This is indeed better language and has been adapted.

5. Please report the value of g_2 used for the images and characterizations in Fig. 1(b,c,e,f) in the caption of Fig. 1.

These are completely fixed molecules ($g_2 = 1$) and this information has been added to the caption.

6. The characterization of the Vortex PSF is overly simplistic in line 140. The Vortex PSF contains both a central spot and a surrounding weak ring that should be mentioned here.

We have adjusted this section to a more detailed and accurate description of the PSF shape. We now describe it as "Now out-of-plane ($\theta = 0$) orientations have constructive interference in the center, generating a central spot surrounded by a dark ring and an additional dim ring. In-plane ($\theta = 90$) orientations have destructive interference in the center resulting in a zero surrounded by a bright ring. Due to the polarization and directional emission from the fixed dipole emitter, the intensity distribution changes along this ring as a function of the azimuthal angle. When varying the polar angle from $\theta=0$ to $\theta=90$ the central bright spot moves outwards asymmetrically, distinctly changing the PSF shape as a function of the polar angle."

7. Please define pixel size (65 nm I believe) when pixel regions of interest are first mentioned in line 189.

We have added the pixel size at this first mention.

8. Please define the meaning of $\pm\sigma$ in the Figure 2 caption. Is this one standard deviation?

This is indeed the case and has been clarified in the caption.

9. Please add additional quantifications of molecule brightnesses:

These additional quantifications have been added.

a. Please add colorbars to Fig. 2(a,c,d) shows image brightnesses in photons/pixel.

Colorbars have been added to all PSF images in Figure 2.

b. Please add information on the average number of photons detected from the molecule

in Fig. 2(a).

This has been added to the caption: "The mean signal count in the frames is 25×10^3 with a standard deviation of 4×10^3 and a mean background of 24 counts per pixel." The standard deviation appears larger than expected, i.e., $\sqrt{N} \sim 158$, because of additional fluctuations in the fluorescence emission. The figure below shows the molecule photon count as a function of the stage position for the estimated count (N) and the sum of pixel values, where both values capture such fluctuation at e.g. $z_{\text{stage}} = -200$. Note that the sum of pixels value appears much larger as it includes both the signal and background (ROI=31x31), whereas the estimated N is the signal only (mean estimated background of $b=24$ photons/pixel).

c. Please provide the average and std. dev. of the number of photons detected needed to achieve the precisions reported in lines 231-232.

The photon counts have been added: "For the 21 molecules, the signal level varied considerably between with 4.4×10^3 and 45×10^3 photons, with a mean of 17×10^3 photons and a standard deviation of 12×10^3 photons"

d. Please report the average and std. dev. of the number of photons detected for molecules 1-3 in the caption of Fig. 3.

This has been added along with the mean and standard deviation of the background: "The photon count per frame (mean \pm one standard deviation) is: $N_1 = 4.7 \times 10^4 \pm 0.2 \times 10^4$ $N_2 = 5.9 \times 10^4 \pm 1.0 \times 10^4$ $N_3 = 3.3 \times 10^4 \pm 0.5 \times 10^4$ and background is: $b_1 = 91 \pm 2.0$ $b_2 = 89 \pm 1.1$ $b_3 = 80 \pm 1.7$ "

e. Please add a colorbar to Fig. 3(a) showing image brightnesses in photons/pixel.

This has been added with a colorbar for each of the 3 different molecules.

f. Please report the average and std. dev. of the number of photons detected for the precisions stated in lines 303-305.

We have added the estimated signal and background photon count distributions of the lambda-DNA experiment to Supplementary Fig. 10 and reported the median values next to the reported precision in the manuscript: "The localization and orientation precision values determined in this way are within the estimated error bars of the CRLB, with the photon distribution showing a median signal of 4600 photons and a median background of 10 photons per pixel (Supplementary Fig. 10)."

10. Please report the use of total internal reflection (TIR) illumination in line 310. This aspect is critical to understanding this experiment and should not be relegated to the Methods section.

This information was indeed missing in this section and has been added: "To that end we have imaged the same region both with and without a quarter waveplate (QWP) in the illumination laser path in Total Internal Reflection Fluorescence (TIRF) conditions (Fig. 5(a))."

11. Both Section 4.1 and Supplementary Note 1 refer to $H(r, \Omega)$, which includes background fluorescence contributions, as a PSF model. However, I believe "PSF" is commonly defined as the response of a microscope to a single point/dipole emitter. Since background must necessarily come from other objects in the sample, it is not correct to refer to $H(r, \Omega)$ as a PSF. It would be better to say "forward model of the imaging system" or simply "image formation model."

We have corrected our terminology to be more consistent with the PSF definition.

12. Line 444: Please clarify what is meant by "fraction of signal photons captured within the ROI..." Does this statement imply that ~55% of the photons emitted by the molecule are simply not collected by the objective lens? Or, does it imply that ~55% of the photons fall outside of a 15x15 pixel region surrounding the image's center of mass? If it is the latter, then this number seems artificially large. Please explain where this calculation comes from, and comment on why the photons are spread so broadly across the detector.

In index matched conditions 86% falls within the ROI as is to be expected from a PSF. However this is calculated in mismatched conditions (the conditions of the Lambda-DNA experiments) with a medium refractive index of 1.333, coverglass index of 1.523 and immersion index of 1.518. This induces a significant amount of spherical aberration that spreads the photons out more than in an ideal case. Therefore this implies that out of all the light captured by the objective ~45% falls within the 15x15 ROI. We have studied the long tails of the PSF and the impact on detected photon count within the ROI before in great detail ([53] Thorsen, R. et al. Impact of optical aberrations on axial position determination by photometry. Nat. Methods 15, 989-990 (2018). <https://doi.org/10.1038/s41592-018-0227-4>), and build on this expertise.

13. Line 592: What are the units for the "constant of around 10"? I assume photons?

The threshold unit is indeed photons and has been added.

14. Supplementary Note 1, line 49: Please define the domain of j . I assume $j=x,y,z$?

This has been added.

15. Supplementary material: I recommend against using i as an index of summation in equation (4), since it is used as an imaginary number of subsequent equations.

This use of i was inconsistent and i as an index of summation has been replaced with h in equation 3, 5 and 6 in Supplementary Note 1.

16. For clarity, please define exactly the aberration function $W(\rho)$ when it is first introduced in line 57 of Supplementary Note 1. I assume it is equal to the sum $K(\rho) + \sum A_n m(x, y) Z_n m(\rho)$?

Your assumption is correct and this has been clarified: "The aberration function $W(\sim\rho) = K(\sim\rho) + W_{abb}(\sim\rho)$ includes the zone function of the vortex phase plate: $K(\sim\rho) = \beta/(2\pi)$ where $\beta = \arctan(\rho_x/\rho_y)$ is the azimuth pupil coordinate, and further includes field-dependent aberrations $W_{abb}(\sim\rho)$ as described in Supplementary Note 2." With $W_{abb}(\sim\rho)$ described in supplementary note 2: "The aberrations function $W_{abb}(\sim\rho)$ is conventionally expressed as a linear sum of the root mean square (RMS) normalized Zernike polynomials $Z_{m n}(\sim\rho)$: $W_{abb}(\sim\rho) = \sum_{n,m} A_{m n} Z_{m n}(\sim\rho)$ "

17. Please adjust the vertical axes of the bias plots in Supplementary Fig. 2 so that these curves are more readable. In particular, the range of g_2 is only -1 to 1, so it is very difficult to read its bias compared to other parameters in degrees in panel (e). The same applies for the intensity bias ΔIII . Perhaps normalizing the bias by the CRB will make these plots more meaningful.

Thank you for the good suggestion. The bias is now shown normalized to the precision to make Supplementary Fig. 2 more readable.

18. For clarity, please replace “unknown” with “unknown (inaccurately calibrated)” in the caption of Supplementary Fig. 3.

This has been clarified.

19. Please report the precise $A_n m(x, y)$ coefficients used for the aberration simulation study in Supplementary Fig. 3.

In all cases $36 m\lambda$ of aberration is added as specified along the y axis.

20. Please comment on the low quality Zernike fittings shown in Supplementary Fig. 4 (h,k,l,n). Why are the reported R^2 values acceptable?

We interpret that the low R-squared value indicates that the field variations are within the estimation precision of those modes. The reviewer refers to modes that are estimated almost constant in the imaging field (now S. Fig. 5), so the proportion of the variance described is within the estimation precision. However, the significant trend is still well-fitted with the surface. To quantify this more clearly in the figure, we have added the root-mean-square error (rmse) to show that it is less than 6 milli-waves for all Zernike maps, and we have kept the R-squared value in the caption.

Reviewer #3 (Remarks to the Author):

In “Simultaneous orientation and 3D localization microscopy with a Vortex point spread function,” Hulleman et al. demonstrate a quantitative imaging method for determining the 3D positions, orientations, and rotational mobilities of single fluorescent molecules. This method can be readily combined with single-molecule super-resolution microscopy experiments to provide complementary biophysical information about the alignment and ordering of the fluorescent probes, which is not usually detected in conventional experiments. The key ingredient of this technique is a vortex phase mask placed at the Fourier (pupil) plane in the emission pathway of a standard fluorescence microscope. This optic modulates the PSF of the imaging system to yield single-molecule images in which orientation and 3D position related effects are more readily apparent, and amenable to image-fitting techniques. The authors convincingly demonstrate their approach using proof-of-concept experiments conducted on single dye molecules in polymers, and intercalating dyes bound to in-vitro Lambda-DNA (intercalators bind to the DNA by sliding between adjacent base pairs, becoming partially rotationally immobilized). The overall precision of the method is benchmarked using Cramer-Rao lower bound calculations. Furthermore, a thorough characterization of field-dependent optical aberrations is performed, and details regarding the phase-mask alignment procedure are provided. These aspects will greatly assist other experimentalists attempting to reproduce results, and use the method for their own work. It is also worth mentioning that the Vortex PSF can be implemented using off-the-shelf components (more traditionally used for STED microscopy). This will potentially facilitate the widespread adoption of the method.

Thank you

Unfortunately, it is my responsibility to point out that many similar methods have very recently been published (Refs. 31 and 32 in the submitted manuscript, as well as Lu et al. doi.org/10.1002/anie.202011444). Those papers, among others, have already convincingly demonstrated the core capabilities of 3D orientation/position imaging and precise rotational mobility characterization in a range of samples. I am concerned that this manuscript may be interpreted as an incremental improvement, if the scope is restricted to methods-development and proof-of-concept. However, due to the otherwise extremely high technical quality and promise of the method, I would strongly advocate that the authors be given the opportunity to revise and further extend their work for publication in Nature Communications. This could involve in-depth analysis of their current dataset and additional experimental measurements. It seems like there is a lot more that could be explored in the intercalated Lambda-DNA example. I have accordingly provided a number of suggestions below for directions that I hope the authors may think worthwhile to pursue. Alternatively, an imaging experiment using rotationally fixed dyes in live cells could also be very compelling and serve to better showcase the advantage of combined 3D orientation + 3D position determination (see for example Mehta et al. doi.org/10.1073/pnas.1607674113).

We thank the reviewer for thinking along with us to pin down better the lambda-DNA experimental results and for suggesting avenues for showcases beyond the proof of principle. We comment on the suggested applications below, but given our limited biological expertise, we sought a collaboration and pursued their application. The group of Cees Dekker (also TU Delft) has a long standing interest and expertise in single molecule biophysics and are as such interested in the imaging of DNA structure/conformation. Together with them and our setup we investigated the supercoiling of DNA (plectonemes). This has resulted in the new section (2.6). On these super-coiled DNA strands we find a different preferential binding orientation of dipoles than for lambda DNA. Additionally we have found that localizations of a certain dipole orientations occur periodically

along the plectoneme. This periodicity of 122-150nm is indicative to the expected pitch of crossings of the super-coiled DNA. In this way the super-coiling can be investigated by light microscopy instead of with e.g. AFM.

Regarding the reference to Lu et al. (<https://doi.org/10.1002/anie.202006207>), this is an application of the tri-spot which we have referred to already as ref [30] (<https://doi.org/10.1063/1.5031759>).

In the authors' Lambda-DNA experiments, they report a difference in delta-phi (azimuthal probe orientation w.r.t. DNA-axis) as a function of different laser excitation polarizations. Notably, it is found that by exciting the sample with significantly z-polarized light, the detected intercalators preferentially assume an orientation of delta-phi ~ 65 deg. (As opposed to 87 degrees when only in-plane illumination is used.) This result is surprising, because it demonstrates there is a significant population of dyes that are not perpendicular to the DNA-axis — which would be highly non-intuitive and add much-needed insight to previous measurements. Furthermore, this result is very interesting because the dyes orient at delta-phi ~65 degrees, but not 115 degrees! As the authors point out, this may be due to the helicity of the DNA structure.

—The asymmetrical preference for delta-phi ~65 over 115 deg. (Fig 5c) is quite remarkable, but this claim really requires more evidence and verification. I would recommend taking data on more DNA aligned along different orientations w.r.t. to the experimental system to try to gauge how reproducible this result actually is. Specifically, if this effect is related to the helicity of the DNA, for some pieces of DNA, one would expect to find a delta-phi distribution centered at 115 deg. INSTEAD of 65 deg.

In the initial submission we had a sentence claiming that the found relative angles were independent of the DNA orientation. To substantiate this claim we have now included the same analysis of 4 additional DNA strands with varying orientations that show essentially the same $\Delta\phi$ - θ distribution (Supplementary figure 11). In some distributions there are less in-plane orientations, this is because the excitation polarization is perpendicular to those dipole orientations and are therefore not excited. Furthermore if the definition of the DNA axis orientation is rotated by 180 degrees the distribution is largely unchanged. In this case to map the azimuthal angle and polar angle back to a range from 0-180 degrees all that happens is an additional mirroring around a polar angle of 90 (for example 270 degrees azimuthal and 45 degrees polar is equivalent to 90 degrees azimuthal and 135 polar angle). As the distribution is largely symmetric around a polar angle of 90 it does not change much to the distribution.

Furthermore, if the authors could figure out a way to determine the corresponding AT-rich and GC-rich regions of a given lambda-DNA, this would provide them the means to correlate the intercalator-tilting direction with the directionality of the helix axis of the DNA.

The idea of identifying AT/GC rich regions to determine the direction is indeed useful to relate the orientations found to the exact DNA structure. Unfortunately there does not appear to be any correlation between localization density and the AT/GC rich regions in our data. With the same sample preparation method and the addition of a high concentration on Sytox Orange appears to yield a uniform intensity along the DNA strand (data not shown). It is possible that the degree of stretching is simply not sufficient to see a preference for GC or AT regions as seen at high degrees of stretching (<https://doi.org/10.1038/s41467-017-02396-1>). The other possibility is that Sytox Orange is less selective for AT/GC rich regions (<https://doi.org/10.7554/eLife.36557>).

Another possibility that the authors should consider is that the observed delta-phi is caused by how the DNA was combed onto the coverslip (i.e. the direction of the receding fluid), and is not related to the directionality of the DNA helix-axis. Helicity effects could be ruled out (or dramatically confirmed) by finding a piece of DNA with an abrupt ~180 degree curve, and ensuring that at both sides of the curve delta-phi is biased in the same direction, despite the fact that the local helix axis would be running in opposite directions, relative to the experimental reference frame.

We have made additional analyses of the orientational parameter distributions of DNA strands with different orientations, shown in Supplementary Figure 11. We found no substantial differences, indicating that the combing direction is not the root cause of the observations.

The other possibility is that the observed tilt could be due to S-DNA formation. This is probably unlikely because the DNA is only slightly extended beyond its contour length, and intercalators will bind to S-DNA-rich regions with much lower affinity. However, I think there are combing protocols that can be used to produce more highly overstretched DNA that could enable a more rigorous study.

We have attempted to generate strands with various degrees of stretching by varying the rotation speed of the spin-coater and performing linear combing at different extraction speeds. Generally it resulted in the longest strands being 5-10% longer. However there is a large variation in the length of strands within one sample. It is unclear if this is a manifestation of varying degrees of stretching or if some of the strands have been cleaved. This unfortunately makes correlating the degree of stretching with the orientational distributions very difficult. From experiments with optical tweezers it appears that this transition would indeed happen at much higher degrees of stretching. In the end we decided not to pursue this and include it into the manuscript as the data was not conclusive.

—The use of excitation polarization to identify the population of dyes with delta-phi close to ~65 deg. is a good idea, but I think this could be extended a bit further. Instead of alternating between horizontal and circularly polarized light, it seems like it would be useful to toggle between pure polarization states (purely in-plane and z-polarized states at the sample). This would allow different populations of dyes to be more cleanly isolated. I bring this up because I think some more investigation could be done about the bias in polar (theta) orientations that the authors observe when a vertical (in the plane of the coverslip) excitation polarization is used. While the authors state in words (line 313) that in-plane excitation leads to theta close to 90 deg., their data (Fig. 5b) seems to suggest that theta has a strong out-of-plane orientation of close to 135 deg. (I am assuming that the vertical axis of the 2D histogram in 5b is polar angle, as in 5c, but the panel is not clearly labeled). When using a linear, in-plane excitation polarization, I wonder how theta changes as a function of the orientation of the DNA-axis? For example, would the detected theta orientation histogram change if the DNA is oriented at 0 versus 45 deg. in the experimental reference frame?

Creating pure polarization states is one of the desires we have, unfortunately the waveplates in the Nikon TIRF illuminator that we use are glued in place allowing no adjustability. Furthermore they are achromatic and not optimal for a specific wavelength. We intend to build a new illumination path in the future to generate and vary pure polarization states to potentially probe the absorption dipole and emission dipole independently. The y axis label of Figure 5 b and d got cut off, this has been fixed.

Without the waveplate the polarization is primarily in-plane and excites molecules with an orientation of theta=90 most effectively (as stated in the original line 313). This however does not imply that this will be the most

occurring orientation, it is just the orientation that is most effectively excited by the excitation laser. The most occurring orientation (or the shape of the orientation histogram) is a property of the fluorophores bound to the lambda-DNA on the cover slip. The excitation polarization only seems to affect which orientations are not effectively excited and therefore missing in the orientation histogram. This can be seen from orientation histograms in Supplementary figure (11 with QWP) of differently oriented DNA strands.

—The use of a symmetric rotational mobility parameter (g_2) is a significant improvement over earlier methods that assume fixed dipole emitters, however this may not be the most appropriate rotational-mobility model for the lambda-DNA sample. In this case, I would expect the intercalated dyes to potentially rotate more about the DNA-axis, while remaining roughly parallel to their neighboring base pairs. From the experimental data that the authors have already collected, they could most likely investigate whether there is any asymmetrical wobble evident in the intercalator fluorescent images. This would require slightly more parameters in their image-formation model, but should be entirely feasible given the sophistication of their current analysis efforts.

We would like to thank the reviewer for his confidence in our analytical capabilities, it is indeed plausible that the orientational potential well in which the intercalators in DNA undergo rotational diffusion does not have full uniaxial symmetry but is perhaps better described as biaxial. The question is if additional orientational parameters can be reliably estimated. To answer this question we must first know how many parameters there are. Following the lines of earlier work from our group (the theory work of ref. 36) one should parameterize the 3×3 matrix R_{ij} found by averaging $d_i d_j$ over the Boltzmann distribution for the orientational potential well, where d_i are the dipole unit vectors. This translates mathematically to figuring out how many parameters are needed to describe a symmetric 2nd rank tensor with trace equal to one. For the uniaxial case (rotational symmetry around the axis corresponding to the potential minimum) that we use the answer is 3: the polar and azimuthal angle of the preferential, minimum energy axis and one anisotropy parameter, the weight factor g_2 . For the biaxial case (no rotational symmetry in the orientational potential well) the answer is 5: we need the three Eulerian angles to characterize a triad of orthonormal vectors d_i , e_i , and f_i for which the matrix R_{ij} is diagonal, and we need two anisotropy parameters (only 2 not 3 because of the trace constraint). This is not inconceivable, but would require a further 2 more parameters to fit, while we already fit 8 (!) parameters (x_0 , y_0 , z_0 , N , b , θ , ϕ , g_2). We think this is asking a bit too much in view of the typical single molecule data quality, but could perhaps be feasible with a more complex method (e.g. add polarization detection). So, we respectfully decline to make the suggested generalization. In the discussion section we now write “The feasibility of estimating the orientational confinement that is not rotationally symmetric around the preferential axis defined by the minimum of the orientational potential well in addition to the other parameters could also be investigated. Going from a uniaxially symmetric to a biaxially symmetric orientational confinement would bring the number of orientational parameters that must be estimated to five. The three Eulerian angles are needed to fix a triad of orthonormal vectors that define the orientation, and two anisotropy parameters are needed to generalize the single parameter g_2 . A reliable estimation of the then total number of 10 parameters may require a more complex setup involving e.g. polarization detection in addition to the vortex phase plate.”

— A more general point the authors may wish to look into would be to contrast their results using a bis-intercalating dye such as YOYO-1. YOYO-1 has two chromophores, and it is generally assumed that each can be modeled as an independent dipole source. However to my knowledge, a precise single-molecule characterization has not yet been done. Using their image-analysis framework, I think it would be extremely compelling if the authors could determine the individual orientations of the two YOYO-1 chromophores within a single dye, as well as their relative tilt angles with respect to the DNA-axis. This type of characterization could potentially help to identify probes that report more information regarding local deformation in the DNA (e.g. changes in the intra-molecular orientation between two intercalated chromophores would be useful for identifying acute changes in DNA structure over ~4 base-pair intervals).

This would indeed be interesting to investigate however the emission of each chromophore is almost impossible to separate from each other as the excitation energy transfers between the chromophores is on the picosecond scale (<https://doi.org/10.1021/ja0609001>). This is much quicker than the fluorescence lifetime ([https://doi.org/10.1562/0031-8655\(2001\)0730585PPOFDD2.0.CO2](https://doi.org/10.1562/0031-8655(2001)0730585PPOFDD2.0.CO2)) and therefore when one chromophore is excited the emission could come from either chromophore. Without any ultra-fast time gating this would always appear as the superposition of two dipole emitters. Under the assumption they have the same azimuthal angle our model could be expanded to a double dipole emitter. To investigate this properly, a phase plate with a design wavelength of 488 or 532 nm should be used and the system should be verified at this wavelength on single molecules. This currently falls outside the scope of this paper. An additional concern is the worse signal-to-background ratio from experiments with YOYO (<https://dx.doi.org/10.1021/nl2025954>).

Signed: ADAM BACKER

REVIEWER COMMENTS

Reviewer #1 (Remarks to the Author):

In this revised version the authors have well addressed my comments and I also think those of the other reviewers. I think the newly added data of plexonemes in DNA is a really nice addition showing the power of the method.

I recommend publication of the manuscript in this version.

Review of *Nature Communications* manuscript NCOMMS-20-43805A

“Simultaneous orientation and 3D localization microscopy with a Vortex point spread function”

In this paper, Hulleman *et al.* demonstrate a vortex point spread function (PSF) for imaging the 3D position, dipole orientation, and degree of rotational constraint of single fluorescent molecules from a single 2D image. The vortex PSF is created using an affordable and commonly available phase plate (used for STED nanoscopy) and has a relatively compact PSF size, making it compatible with existing labeling techniques utilized in single-molecule localization microscopy (SMLM). The authors show that the vortex PSF is effective at measuring ATTO dyes within a polymer (PMMA) layer, re-orientation dynamics of ATTO dyes on glass, Sytox orange intercalated within elongated lambda-DNA, and the structure of supercoiled DNA, as long as the signal-to-background ratio is relatively strong (≥ 2000 signal photons and $SBR \geq 40$). The authors also carefully report the instrument alignment and calibrations, especially in terms of optical aberrations and instrument drift, that are necessary to enable optimal imaging performance.

Overall, I applaud the authors for making substantial and helpful improvements, thereby addressing my previous concerns in this revised manuscript. In particular, the quantitative analyses are of top quality that enable readers to fully appreciate and evaluate the power of the proposed imaging technique. The authors include ample evidence that supports the future potential of the vortex PSF for orientation and 3D localization microscopy. Importantly, the work is of high significance for both the novelty and ease of implementation of the technique, as well as the new quantification of the behavior of DNA intercalators.

I recommend that the manuscript be accepted for publication in *Nature Communications*, subject to the minor suggestions for revision listed below.

1. It is generally difficult to read quantitatively the orientation distributions depicted in the hemispherical histograms in Figs. 5(b,c), 6(b), etc.
 - a. I suggest that the authors add latitude and longitude tick marks and labels to orient the reader to the coordinate system.
 - b. Please also report the bin size used in these histograms – I assume that they are of uniform solid angle?
 - c. The included colorbars are a bit difficult to interpret because 1 sr is much much larger than the size of each bin. Can the authors instead simply use units of counts/bin?
2. Please enlarge the font size of the legends in Fig. 6 and similar figures. They are much too small to read at normal zoom levels.
3. Line 452, main text: I found it difficult to locate the mentioned rotationally symmetric population in the data. Please reference a figure here. I assume the authors are referring to the population shown in Supplementary Fig. 16(b)? Additional annotations (e.g., arrows) within the relevant figures could also be helpful.

4. Fig. 6(e) shows an autocorrelation of data that I cannot quite find in the manuscript. The ~ 150 nm periodicity in localization density is not apparent in the orientation histogram in Fig. 6(c) or the scatterplot in Fig. 6(d). Please explicitly include a plot of the binned localization density data used in Fig. 6(e) vs. Supplementary Fig. 14.
5. While the localization data of Sytox orange are generally of high quality, it is a bit difficult to judge the statistical power of the autocorrelation analyses. Please include a table that summarizes the total number of localizations and effective linear localization density (in localizations/nm) along the DNA for the various DNA data.
6. Supplementary Note 1, line 53:
 - a. Rotational diffusion coefficients typically do not have units of time and thus are difficult to compare directly to fluorescence lifetime. Perhaps the authors mean to say “rotational correlation time” here instead?
 - b. I believe “fluorescence lifetime” should be used here instead of “fluorescent lifetime.”
7. A final observation: Supplementary Fig. 2(e,f) suggests that the MLE implemented in this work has difficulty discerning signal from background at low SBRs (exemplified by the anti-correlated bias curves of I and b). It is possible that this bias contributes to bias in estimating g_2 , but overestimates of g_2 at low SBRs could also be a symptom of severe shot noise (discussed in Ref. [50]).

It is possible that the authors' choice of using small ROIs to simultaneously fit signal N and background b contributes to this difficulty. Since fluorescence background is generally smoothly correlated across the entire field of view, it may be beneficial to use separate large ROIs in space (and perhaps time) to robustly estimate b before attempting to measure SM locations and orientations within each small ROI. This strategy could enable the vortex PSF to be used at lower SBRs.

Reviewer #3 (Remarks to the Author):

In the revised manuscript "Simultaneous orientation and 3D localization microscopy with a Vortex point spread function" Hulleman et al. have addressed my greatest concern with the earlier submission by providing additional experimental results to further showcase the capabilities of their imaging method. I am very happy to see that the authors have used their method to shed new insight on a supercoiled lambda-DNA sample. I would like to recommend publication provided the authors have satisfied the other reviewers' criticisms. However I have some additional comments below:

Comments on the results in Fig. 5/ bias in $\Delta\phi$ when using polarized illumination: I appreciate that care has been taken to verify the measurements using DNA strands at different orientations with respect to the microscope coverslip. However there are a few things that remain a bit confusing to me. Please label in Fig. 5(a) the orientation of the in-plane component of the excitation polarization. In Fig. 5(b,c) it would be helpful to clearly note in the figure the orientation of the DNA axis (the x-axis?) as well as the orientations of the excitation polarizations with and without the QWP (When using the QWP, can the authors provide a rough estimate of the relative magnitudes of the in-plane and out-of-plane excitation polarization components?). I would also like the authors to highlight (as was done in Fig. 6(b)) the portions of the density map that contribute to an overall $\Delta\phi$ of 65 degrees. Inspecting 5(c) by eye, the bias/asymmetry seems less pronounced, so maybe a few ticks added onto the figure with degrees labeled would be helpful. Overall, this is a novel observation that could merit future study.

Comments on Fig. 6 / Plectoneme results: The new data included in Fig. 6 provides an excellent example of the combined benefits of super-resolved (spatial) imaging with single-molecule orientation measurement. Interestingly, the authors use their method to reveal a periodic feature (~ 130 nm) in supercoiled, torsionally constrained DNA, which is most likely due to the DNA wrapping around itself. The result is impressive, but I would like the following questions answered in the final manuscript:

- 1) Could the authors estimate the overall contour length of the DNA in the plectoneme constructs? How does this compare to the contour length in the (torsionally unconstrained) lambda-DNA samples prepared by spincoating? Does the new sample preparation and different attachment method provide a means of generating a more highly overstretched construct?
- 2) It is interesting to see that in the regions of the DNA where the strands are not twisted around each other, the dyes do not orient perpendicular to the DNA axis (Fig. 6b). Some further discussion/speculation here might be useful. It could be helpful to additionally compare the overall localization density in the non-overlapping region to a region where the strands are twisted around each other. If the localization density is significantly less than 50% that in the overlapping region, these combined observations could potentially be evidence of a different DNA conformation being present (e.g. S-DNA).
- 3) The authors reveal ~ 130 nm periodicity in the twisted region of the construct alternately by selecting a subset of molecules binned in a particular range of orientations, and computing the autocorrelation of the localization density. Could the authors provide a brief explanation of how this subset of orientations was chosen, and what proportion of detected molecules fall into the range of "useful" orientations that help to reveal the underlying periodicity? If a different subset of orientations were chosen to analyze, would a periodic structure still be recognizable (perhaps with some phase-shift relative to the one shown in the figure)?

I have one more reply regarding the authors' rebuttal to my previous criticisms regarding a symmetric rotational mobility fitting parameter and use of a bis-intercalating dye: The authors imply that the increased number of parameters will make fitting image data to an asymmetrical potential well infeasible. I don't think this additional fitting is as difficult as they claim, since it basically just amounts to fitting three orthogonal superimposed dipole images of varying intensities. If the increased number of fitting parameters is really a major concern, then a uniaxial model could be considered in which there is rotational symmetry about the potential maximum as opposed to the minimum. The key thing to investigate would be whether there is any evidence

that the intercalators enjoy more rotational freedom in the plane perpendicular to the DNA axis. A similar analysis approach could be taken to investigate a bis-intercalator. Although the image sensor cannot temporally resolve energy transfer between the chromophores, one could imagine fitting two superimposed dipole images to the raw image data in order to estimate the angle between the two dipoles. Although I think both of these points could merit further investigation, I agree that this may fall outside of the scope of the current work.

We thank the reviewers for carefully considering our revisions. We have addressed their remaining concerns below. Besides changes to the text quoted below, there are two changes to the manuscript. Firstly there are additional indicators and a grid for the hemispherical histogram to ease the reader's interpretation of these figures. Secondly we have added two Supplementary tables (1&2), the original Supplementary table 1 has now become Supplementary table 3. The first table quantifies the amount of localizations and localization density of analyzed strands in various figures. The second table shows the localization density on various sections of the plectoneme construct. Additionally we have added a reference to describe the synthesis and purification of coilable 42kb DNA constructs in more detail (<https://doi.org/10.1101/2021.05.15.444164>).

Detailed response to the reviewers:

REVIEWER COMMENTS

Reviewer #1 (Remarks to the Author):

In this revised version the authors have well addressed my comments and I also think those of the other reviewers. I think the newly added data of plexonemes in DNA is a really nice addition showing the power of the method.

I recommend publication of the manuscript in this version.

Thank you.

Reviewer #2 (Remarks to the Author):

In this paper, Hulleman et al. demonstrate a vortex point spread function (PSF) for imaging the 3D position, dipole orientation, and degree of rotational constraint of single fluorescent molecules from a single 2D image. The vortex PSF is created using an affordable and commonly available phase plate (used for STED nanoscopy) and has a relatively compact PSF size, making it compatible with existing labeling techniques utilized in single-molecule localization microscopy (SMLM). The authors show that the vortex PSF is effective at measuring ATTO dyes within a polymer (PMMA) layer, re-orientation dynamics of ATTO dyes on glass, Sytox orange intercalated within elongated lambda-DNA, and the structure of supercoiled DNA, as long as the signal-to-background ratio is relatively strong (≥ 2000 signal photons and $SBR \geq 40$). The authors also carefully report the instrument alignment and calibrations, especially in terms of optical aberrations and instrument drift, that are necessary to enable optimal imaging performance.

Overall, I applaud the authors for making substantial and helpful improvements, thereby addressing my previous concerns in this revised manuscript. In particular, the quantitative analyses are of top quality that enable readers to fully appreciate and evaluate the power of the proposed imaging technique. The authors include ample evidence that supports the future potential of the vortex PSF for orientation and 3D localization microscopy.

Importantly, the work is of high significance for both the novelty and ease of implementation of the technique, as well as the new quantification of the behavior of DNA intercalators.

I recommend that the manuscript be accepted for publication in Nature Communications, subject to the minor suggestions for revision listed below.

Thank you.

1. It is generally difficult to read quantitatively the orientation distributions depicted in the hemispherical histograms in Figs. 5(b,c), 6(b), etc.

a. I suggest that the authors add latitude and longitude tick marks and labels to orient the reader to the coordinate system.

Done. Furthermore key orientations are highlighted which should help orient readers to the coordinate system.

b. Please also report the bin size used in these histograms – I assume that they are of uniform solid angle?

Bins of uniform solid angle would be ideal, this is, however, quite tedious to implement. The bin size used is 5 degrees in the polar angle and 10 degrees in the azimuthal angle. This has been added to all relevant captions. We think that even this sub-optimal solution for visualization allows the reader to appreciate the finding.

c. The included colorbars are a bit difficult to interpret because 1 sr is much larger than the size of each bin. Can the authors instead simply use units of counts/bin?

If the bins were of uniform solid angle then counts/bin would be the logical choice. For a fair representation with non-uniform bins (see above) we have normalized the counts in each bin to the size of the bin.

2. Please enlarge the font size of the legends in Fig. 6 and similar figures. They are much too small to read at normal zoom levels.

Done. (also in Supplementary Fig. 13-15).

3. Line 452, main text: I found it difficult to locate the mentioned rotationally symmetric population in the data. Please reference a figure here. I assume the authors are referring to the population shown in Supplementary Fig. 16(b)? Additional annotations (e.g., arrows) within the relevant figures could also be helpful.

The population mentioned in line 452 is indeed not visible in any of the presented orientation sphere's. These populations are present on other DNA strands that are presumably oriented in the opposite direction. This has been rephrased to: "... occurs on other DNA strands...(data not shown)."

The population shown in Supplementary Fig. 16(b) is not 180 degree symmetric and is referred to later around line 469: "Lastly the orientation of supercoiled sections before and after the plectoneme appear to have opposing orientation shifts. The peak relative orientation in the region to the top right of a plectoneme is

$\Delta\phi = 111^\circ$ $\vartheta = 36^\circ$ and to the bottom right is $\Delta\phi = -102^\circ$ $\vartheta = 29^\circ$, with the full distribution of all 3 section of a plectoneme shown in Supplementary Fig. 16. These two orientations appear shifted in opposite directions and are not 180 degree rotation symmetric which would be expected from a non-supercoiled strand.” As described in the text there appears to be an opposing orientation shift between the upper section before super-coiling and the lower section. In Supplementary Fig. 16 (b-c) we have added arrows pointing to the average value around the peak. In Fig. 5 (b-c) we have added dashed lines highlighting constant azimuthal angles.

4. Fig. 6(e) shows an autocorrelation of data that I cannot quite find in the manuscript. The ~150 nm periodicity in localization density is not apparent in the orientation histogram in Fig. 6(c) or the scatterplot in Fig. 6(d). Please explicitly include a plot of the binned localization density data used in Fig. 6(e) vs. Supplementary Fig. 14.

Fig 6(e) (now Fig 6(f)) shows the autocorrelation of binned localizations from the light blue data in figure 6(d). The bottom panel of the original Fig. 6(d) has been replaced with a new panel named Fig. 6(e) that shows the binned localization density from the region displayed in Fig. 6(d) and a section of similar length from Supplementary Fig. 14(b).

The text regarding this section has been changed to: “The localizations are binned in 6.5-nm intervals along the x-direction, such that an autocorrelation of the localization density can be calculated. The periodic pattern from Fig. 6(d) is more evident in the binned localizations in Fig. 6(e), compared to binned localizations from a torsionally relaxed DNA molecule (from Supplementary Fig.~14(b)) where no periodicity can be clearly identified.”

5. While the localization data of Sytox orange are generally of high quality, it is a bit difficult to judge the statistical power of the autocorrelation analyses. Please include a table that summarizes the total number of localizations and effective linear localization density (in localizations/nm) along the DNA for the various DNA data.

We have added a table (Supplementary Table 1) to summarize the localizations analyzed for most figures. This table includes the total number of localizations along the DNA strand, number of localizations in the specified orientational subset, length, and localization density. The localization density is calculated for the analyzed orientational subset unless in the corresponding figure no orientational subset is analyzed.

6. Supplementary Note 1, line 53:

a. Rotational diffusion coefficients typically do not have units of time and thus are difficult to compare directly to fluorescence lifetime. Perhaps the authors mean to say “rotational correlation time” here instead?

The units are indeed inconsistent and hard to compare. We changed it to “rotational relaxation time”

b. I believe “fluorescence lifetime” should be used here instead of “fluorescent lifetime.”

We thank the reviewer for catching this mistake.

7. A final observation: Supplementary Fig. 2(e,f) suggests that the MLE implemented in this work has difficulty discerning signal from background at low SBRs (exemplified by the anti-correlated bias curves of l and b). It is possible that this bias contributes to bias in estimating g_2 , but overestimates of g_2 at low SBRs could also be a symptom of severe shot noise (discussed in Ref. [50]).

The anti-correlated bias curves of l and b do indeed suggest that at low SBR it is harder to discern these two. This effect is known from fitting of freely rotating emitters already [see ref 55 Supp. Fig 1]. This bias is reduced for larger ROI but this is slower to process and might not be applicable for dense samples. In general it is true that at low SBR the g_2 estimate becomes unreliable due to shot noise. This could both under estimate the rotational diffusion of fixed emitters and overestimate the diffusion of freely rotating emitters.

We have adjusted and added the following to Supplementary Fig. 2:

"The slight bias in g_2 is primarily caused by the limited estimation range $[0, 1]$. At very low SBR, rotational diffusion estimates can be inaccurate due to shot noise [4].

It is possible that the authors' choice of using small ROIs to simultaneously fit signal N and background b contributes to this difficulty. Since fluorescence background is generally smoothly correlated across the entire field of view, it may be beneficial to use separate large ROIs in space (and perhaps time) to robustly estimate b before attempting to measure SM locations and orientations within each small ROI. This strategy could enable the vortex PSF to be used at lower SBRs.

Increasing the ROI (assuming there is enough sparsity) would help indeed to discern signal N from background b , especially at low SBR. The strategy to estimate the background not at the same location as the emitter fit seems problematic in general. We think this only shifts the problem, and might result in inaccurate background estimation. From an estimation point of view a large ROI might be best, even though then the background might not be constant anymore, but from a sparsity point of view this is very limiting in the experiment.

Reviewer #3 (Remarks to the Author):

In the revised manuscript "Simultaneous orientation and 3D localization microscopy with a Vortex point spread function" Hulleman et al. have addressed my greatest concern with the earlier submission by providing additional experimental results to further showcase the capabilities of their imaging method. I am very happy to see that the authors have used their method to shed new insight on a supercoiled lambda-DNA sample. I would like to recommend publication provided the authors have satisfied the other reviewers' criticisms. However I have some additional comments below:

Thank you.

Comments on the results in Fig. 5/ bias in $\Delta\phi$ when using polarized illumination: I appreciate that care has been taken to verify the measurements using DNA strands at different orientations with respect to the microscope coverslip. However there are a few things that remain a bit confusing to me. Please label in Fig. 5(a) the orientation of the in-plane component of the excitation polarization. In Fig. 5(b,c) it would be helpful to clearly note in the figure the orientation of the DNA axis (the x-axis?) as well as the orientations of the excitation polarizations with and without the QWP (When using the QWP, can the authors provide a rough estimate of the relative magnitudes of the in-plane and out-of-plane excitation polarization components?). I would also like the authors to highlight (as was done in Fig. 6(b)) the portions of the density map that contribute to an overall $\Delta\phi$ of 65 degrees. Inspecting 5(c) by eye, the bias/asymmetry seems less pronounced, so maybe a few ticks added onto the figure with degrees labeled would be helpful. Overall, this is a novel observation that could merit future study.

The estimated polarizations was mentioned in the previous rebuttal (without QWP 240:1 $\phi=90^\circ$ $\vartheta=90^\circ$, with a QWP 3:1 $\phi=90^\circ$ $\vartheta=15^\circ$). In the main text the simplification "With a QWP the polarization is half in-plane and half out-of-plane resulting in less selective excitation." was used to help the interpretation of the results as the exact polarization does not have a huge influence. The polarization of excitation only influences which emitter orientations are not excited and thus not localized. In figure 5(a-c) red arrows now indicate the applied excitation polarization. The polarization extinction ratio and primary orientation are now also included in the manuscript text: "Without the QWP the excitation polarization is s-polarized (polarization extinction ratio of 240:1 and an excitation polarization orientation of $\vartheta = 90^\circ$ and $\phi = 90^\circ$), most effectively exciting molecules around $\vartheta = 90^\circ$ and $\phi = 90^\circ$ (Fig. 5(b)). With a QWP the polarization is less in-plane and more out-of-plane resulting in less selective excitation (polarization extinction ratio of 3:1 and a primary excitation polarization orientation of $\vartheta = 15^\circ$ and $\phi = 90^\circ$)." In Fig 5 (b-c) we have added a dashed curve to highlight constant azimuthal angles ($\Delta\phi = 84$ degrees and $\Delta\phi = 65$ degrees respectively). Furthermore we have added a coarse grid with 30 degrees spacing to help interpret these histograms.

Comments on Fig. 6 / Plectoneme results: The new data included in Fig. 6 provides an excellent example of the combined benefits of super-resolved (spatial) imaging with single-molecule orientation measurement. Interestingly, the authors use their method to reveal a periodic feature (~130 nm) in supercoiled, torsionally constrained DNA, which is most likely due to the DNA wrapping around itself. The result is impressive, but I would like the following questions answered in the final manuscript:

1) Could the authors estimate the overall contour length of the DNA in the plectoneme constructs? How does this compare to the contour length in the (torsionally unconstrained) lambda-DNA samples prepared by spincoating? Does the new sample

preparation and different attachment method provide a means of generating a more highly overstretched construct?

Based on the reviewer's suggestion we have estimated the overall contour length based on the localizations of the plectoneme construct (Supplementary Table 2). This length is 10% shorter than the crystallographic length, which could be because small deviations from the central axis in the intertwined DNA are not discernable in the localization data leading to an underestimate of the actual DNA length. Therefore it is difficult to assess the degree of stretching (if present anyway) reliably. This has been added in the manuscript: "The length of the supercoiled DNA molecule estimated from localizations is 12.6 +/- 0.4 micrometer (Supplementary Table~2). Compared to the crystallographic length the estimated length is 10% shorter, which may be attributed to the intertwining of the DNA molecule as the straight line projection is an underestimate of the actual DNA molecule length."

2) It is interesting to see that in the regions of the DNA where the strands are not twisted around each other, the dyes do not orient perpendicular to the DNA axis (Fig. 6b). Some further discussion/speculation here might be useful. It could be helpful to additionally compare the overall localization density in the non-overlapping region to a region where the strands are twisted around each other. If the localization density is significantly less than 50% that in the overlapping region, these combined observations could potentially be evidence of a different DNA conformation being present (e.g. S-DNA).

The different orientation of the fluorescent dipoles in this sample could be due to the different surface preparation and attachment method. This would be similar to the reasoning why there is a correlation between the polar and azimuthal angle in the Lambda-DNA sample. Some speculation about this observation is added in the manuscript: "This different azimuthal orientation found in the supercoiled DNA sample could be due to the different surface attachment protocol. This could change the physical space or electrostatic potential around the intercalator and DNA molecule."

The localization density in the region where the strands are twisted around each other is approximately 2 times the localization density of the individual sections before crossing over. For solid conclusions about a change in localization density, a systematic study on many more plectonemes needs to be carried out. To address this point in the manuscript we have added: "The ratio between localization density on the coiled section and the individual strands before crossover is 1.94 +/- 0.15 (Supplementary Table~2). This doubling of the localization density confirms there are two DNA strands in the coiled section. A significant change in the binding affinity in either of the sections could indicate a change in DNA conformation [42]. In future, it will be of interest to carry out a systematic study on many more plectonemes."

3) The authors reveal ~130 nm periodicity in the twisted region of the construct alternately by selecting a subset of molecules binned in a particular range of orientations, and

computing the autocorrelation of the localization density. Could the authors provide a brief explanation of how this subset of orientations was chosen, and what proportion of detected molecules fall into the range of "useful" orientations that help to reveal the underlying periodicity? If a different subset of orientations were chosen to analyze, would a periodic structure still be recognizable (perhaps with some phase-shift relative to the one shown in the figure)?

As mentioned, this subset is chosen because it is one of the primary orientations found on individual strands before the start of the super-coiled section. The proportion of detected molecules in each subset is now included in Supplementary table 1 for most individually analyzed DNA strands. The subset analyzed here is about 10% of the complete set of localizations. The periodicity is generally recognizable in a range of $\Delta\phi = 88^\circ$ to $\Delta\phi = 148^\circ$, which suggests that ~15-20% of localizations can be used to identify the periodicity.

On a limited few other orientational subsets a periodicity twice as long was found. Analyzing a cross correlation between subsets close to and the actual subset centered around $\Delta\phi=118^\circ$ and $\vartheta=32^\circ$, a shift of the central peak can be found on two out of three plectonemes that show a shift of 65 nm / 90 degrees and 25 nm / 40 degrees. Some of the peaks identified in the orientational subset also appear on a few of the full-data autocorrelations. This is to be expected as it is one of the primary orientations of the individual strands before intertwining itself. However, from the full-data autocorrelations there are too many other aperiodic peaks that it is difficult to identify the periodicity in this full-data.

We added the following to clarify these points: "..., representing ~10% of all localizations" and "A few of the periodic peaks found in these orientational subsets can be identified in the full-data autocorrelation but cannot be directly identified as there are too many aperiodic peaks in the full-data autocorrelation."

The additional Supplementary Table 1 is already referred to earlier in the manuscript.

I have one more reply regarding the authors' rebuttal to my previous criticisms regarding a symmetric rotational mobility fitting parameter and use of a bis-intercalating dye: The authors imply that the increased number of parameters will make fitting image data to an asymmetrical potential well infeasible. I don't think this additional fitting is as difficult as they claim, since it basically just amounts to fitting three orthogonal superimposed dipole images of varying intensities. If the increased number of fitting parameters is really a major concern, then a uniaxial model could be considered in which there is rotational symmetry about the potential maximum as opposed to the minimum. The key thing to investigate would be whether there is any evidence that the intercalators enjoy more rotational freedom in the plane perpendicular to the DNA axis. A similar analysis approach could be taken to investigate a bis-intercalator. Although the image sensor cannot temporally resolve energy transfer between the chromophores, one could imagine fitting two superimposed dipole images to the raw image data in order to estimate the angle

between the two dipoles. Although I think both of these points could merit further investigation, I agree that this may fall outside of the scope of the current work.

As mentioned the addition of polarization detection will help to analyze the asymmetry in rotational diffusion. With simulated asymmetric rotational diffusion Vortex PSF's the g_2 coefficient is primarily determined by the diffusion in the polar direction. This is to be expected as the Vortex PSF's polar precision is better than the azimuthal precision. By adding polarized excitation or polarization split emission, extra information about the in-plane diffusion is revealed through the modulation contrast or intensity ratio between the two channels respectively. With this additional channel of information estimating all these parameters seems feasible. Without this extra information there will only be very small variations in the Vortex PSF shape, therefore extracting an in-plane and out-of-plane rotational diffusion coefficient would only be feasible with very high SBR.

The idea of investigating bis-intercalators does remain intriguing and a double dipole model, possibly without rotational diffusion does seem plausible. This idea has been added to the section about possible future studies: "Another intriguing possibility is to study the characteristics of bis-intercalators with a double-dipole model."

REVIEWERS' COMMENTS

Reviewer #3 (Remarks to the Author):

The authors have addressed all my criticisms and I am happy to recommend publication. I enjoyed reading the revised manuscript.